# Skilled motor control of an inverted pendulum implies low entropy of states but high entropy of actions

**Nicola Catenacci Volpi**[1]*, **Martin Greaves**[1], **Dari Trendafilov**[2], **Christoph Salge**[1], **Giovanni Pezzulo**[3], **Daniel Polani**[1]

1 Department of Computer Science, University of Hertfordshire, Hatfield, England, United Kingdom, 2 Institute for Pervasive Computing, Johannes Kepler University, Linz, Austria, 3 Institute of Cognitive Sciences and Technologies, National Research Council, Rome, Italy

* n.catenacci-volpi@herts.ac.uk

**Data Availability Statement:** All relevant data are included in the paper's Supporting information files. The source code and data used to produce the results and analyses presented in this

## Abstract

The mastery of skills, such as balancing an inverted pendulum, implies a very accurate control of movements to achieve the task goals. Traditional accounts of skilled action control that focus on either routinization or perceptual control make opposite predictions about the ways we achieve mastery. The notion of routinization emphasizes the decrease of the variance of our actions, whereas the notion of perceptual control emphasizes the decrease of the variance of the states we visit, but not of the actions we execute. Here, we studied how participants managed control tasks of varying levels of difficulty, which consisted of controlling inverted pendulums of different lengths. We used information-theoretic measures to compare the predictions of alternative accounts that focus on routinization and perceptual control, respectively. Our results indicate that the successful performance of the control task strongly correlates with the decrease of state variability and the increase of action variability. As postulated by perceptual control theory, the mastery of skilled pendulum control consists in achieving stable control of goals by flexible means.

## Author summary

What characterises skilled behaviour? When one observes capable tennis players or other masters of their craft, their performance appears effortless, their behaviours purposeful and stable. This has been suggested in some traditional accounts to imply that mastery of a task is linked to highly routinised behaviours; such behaviours would concentrate on well-rehearsed routes. Others have proposed instead that mastery consists in keeping the states well confined during a behaviour; where necessary, actions would vary strongly to achieve that. Which of these is the case? Here, we undertake a study where participants have the task to balance an inverted pole. The length of the pole can be varied, which permitted us to control the difficulty of the task. Using methods from information theory, we set out to answer above question in the case of pole balancing control. We found that successful balancing correlates strongly with focused, well controlled states, but at the cost of

manuscript are available on GitLab at https://gitlab.com/uh-adapsys/pendulum-experiment.

**Funding:** This research received funding from the European Commission, which awarded D.P. as part of the CORBYS (Cognitive Control Framework for Robotic Systems) project under contract FP7 ICT-270219 (www.corbys.eu); from the European Commission's Horizon 2020 Framework Programme for Research and Innovation under the Specific Grant Agreement No. 945539 (Human Brain Project SGA3) to G.P.; and from the European Research Council under the Grant Agreement No. 820213 (ThinkAhead) to G.P. The funders had no role in the study design, data collection and analysis, decision to publish, or preparation of the manuscript.

**Competing interests:** The authors have declared that no competing interests exist.

having highly varying actions. In other words, a potentially wide choice of actions serves to maintain the system tightly in preferred states. According to these results, mastery (in this task) is not about choosing the same actions repeatably, but about reliably achieving the same outcomes.

## 1 Introduction

Humans are able to learn sophisticated skills [1], such as playing a musical instrument professionally, excelling in complex sports such as tennis, balancing an inverted pendulum, or driving a car in a busy city. Yet, the informational principles underlying human skilled action control are incompletely known [2–7].

Classical theories of skill learning in motor control assume that with training actions become routinized, which implies a drastic reduction of their variability [8]. Action variability might be initially useful during learning to promote exploration and remains also afterwards, but in general, a hallmark of expertise and routinization is the low variability of skilled actions [9–13]. The idea that with expertise action variability decreases is also implicit in standard machine learning and AI approaches to gaming (e.g., deep RL systems), where the goal is to learn a convenient policy, or state-action mapping [14, 15]. The massive learning process of deep RL systems implies a routinization of actions and the drastic reduction of the variability of the selected policies—in the sense that at the end of learning, a single policy is retained from a potentially huge space of possible policies considered before learning.

Other frameworks assume instead that we control (and reduce the variability of) perceptual states, not of actions. A prominent example of this perspective is *perceptual control theory* (PCT), which assumes that behaviour uses negative feedback control loops [16–18]. This feedback control loop maintains a controlled perceptual variable at (or close to) a goal or reference point, by producing appropriate actions that—via the environment—influence the controlled variable. For example, when we drive a car, we could keep the speed indicator fixed on our desired speed (say, 80 mph). To do this, we could use an internal comparator to measure differences between the currently sensed input and the reference—and accelerate and decelerate as needed, to reduce this discrepancy. In turn, acceleration and deceleration have an effect on the car dynamics and ultimately on the input that we sense, creating a feedback loop that affords control despite external disturbances (e.g., wind). This example illustrates an important insight of PCT: the fact that the controlled variable is not the output of the system (the actions) but its input. As a consequence, we vary and allow for a potentially high variability in our actions (e.g., press or release break) in order to control our perceptual variable (e.g., speed indicator), which therefore shows much less variability. Importantly, the feedback control loop of PCT can be extended hierarchically, by allowing hierarchically higher levels to set the goals or reference points of hierarchically lower levels and the lower levels to achieve these goals set by higher levels. Most sophisticated behaviours can be described in terms of a PCT controller with multiple hierarchical levels. For example, Johnson et al realized a 4-level perceptual controller for an inverted pendulum that is especially effective in withstanding external perturbations [19]. Other related approaches, such as planning-as-inference, active inference, surprise-minimising RL and KL control are also based on the idea that goal-directed behavior amounts to reducing the entropy or variance of the final (goal) state(s) of the controlled dynamical system [20–27]. For example, when balancing an inverted pendulum, the task ends in one single state, which is the one with the pendulum upwards and standing still. In most practical cases, it is important to reduce the entropy not just of the final (goal) state, but also of (some of) the

intermediate states that the agent has to visit to achieve its goals. This is because in difficult control tasks, such as balancing an inverted pendulum, there is only a "narrow corridor" in the phase space of the task: only a very small subset of trajectories (compared to the full set of possible trajectories in the state space) afford goal achievement. In other words, a difficult control task implies that only a small region of the state space must be visited with high probability to reach the goal. Some exploration of the state space (here, indexed by high state entropy) might be useful during initial phases of learning, or when the control task is easy, but could otherwise hinder the performance. Please note that in this paper, we focus on control dynamics *after* the learning stage and hence we do not consider the effects of learning.

We designed an experiment to adjudicate between these contrasting views of expertise in the case of a pendulum balancing task—and to assess the relative importance of reducing variability of actions and states during skilled control for this problem. We analyzed human participants' behaviour and performance during the control of an inverted pendulum. Crucially, in different trials, the pendulum length varied, therefore creating control tasks of different levels of difficulty [28, 29]. We reasoned that if, in the case of a pole balancing task, the former hypothesis (reduction of action variability) is correct, the skilled control of a pendulum should be indexed by low levels of action entropy. Rather, if the second hypothesis (reduction of perceptual state or outcome variability) is correct, skilled control of a pendulum should be indexed by low levels of state entropy, but not action entropy. To preview our results, we found that successful performance in pole balancing strongly correlates with the decrease of state variability and the increase of action variability. A numerical study based on Markov Decision Processes [14, 15] suggests an important role of noise in determining the trade-off between performance and information processing that we observed in human participants.

## 2 The experiment

We studied the behaviour of 36 human participants performing a continuous control task, consisting in inverting and balancing a swinging pendulum about its axis for as long as possible, using a computer simulation.

The participants of the experiment were recruited from the students and staff of the University of Hertfordshire. They were aged between 22 and 51 years, with 9 female and 27 male participants. Participants had no prior knowledge of the project or the specific task being employed for testing. The experiment was approved by the Ethics Committee of the University of Hertfordshire as complying with the University's policies regarding studies involving the use of human participants (study approved by University of Hertfordshire Science and Technology Ethics Committee and Designated Authority with Protocol No. COM SF UH 00016). Informed written consent was provided by all participants.

### 2.1 Controlled system

The non-linear dynamical system controlled by the participants is a pendulum composed by a mass $m$ hanging from a weightless rod of length $L$ (see Fig 1). We denote with $\theta_t$ the angle that the pendulum has with its vertical axis at time $t$, where $\theta_t = 0$ rad indicates the downward position. In the experiment we set $m = 1$ kg and $g = 1$ ms$^{-2}$ for the gravitational acceleration $g$. Hence, the dynamics of the controlled pendulum is governed by the following differential equation

$$\ddot{\theta}_t \doteq \frac{d^2\theta_t}{dt^2} = \frac{u_t - \sin\theta_t}{L} \tag{1}$$

where $u_t$ represents the contribution of the mass acceleration controlled by the participants at

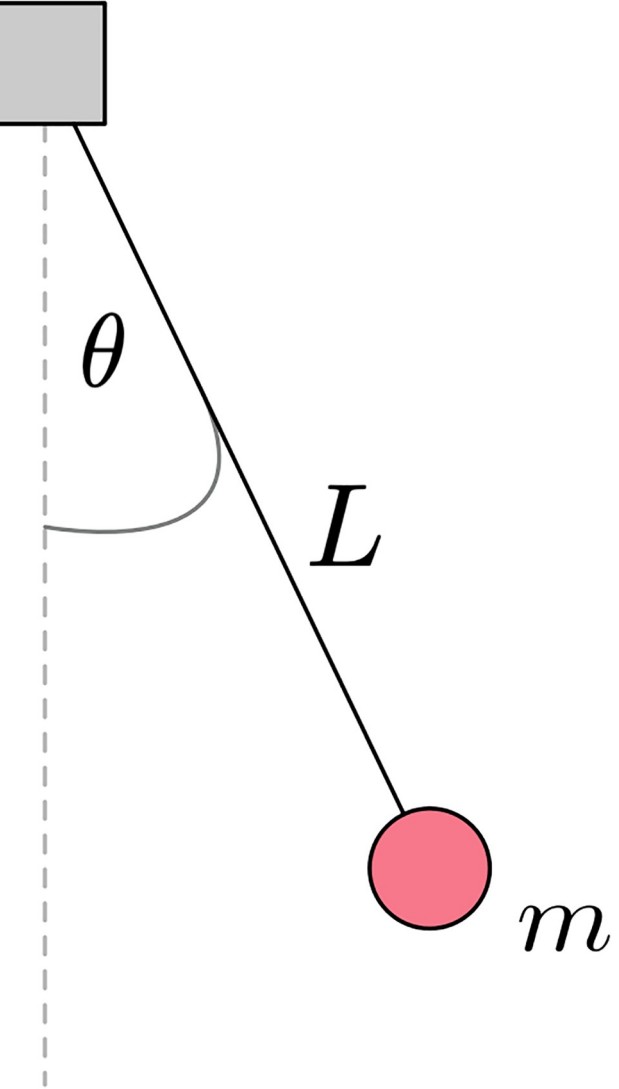

**Fig 1. The controlled system is a pendulum with mass _m_ hanging from a weightless rod of length _L_.** The angle between the pendulum and its vertical axis is denoted by $\theta$, where $\theta = 0$ rad indicates the pendulum's downward position. The goal of the task assigned to participants is to swing the pendulum in order to balance it in its upright position for as long as possible (i.e., $\theta \simeq \pi$ rad and $\dot{\theta} \ll 1$ rad s$^{-1}$).

time _t_. In the simulation this equation is approximated using the Euler method for numerical integration. Given a time interval $\Delta t$, in the numerical simulation the pendulum's angular speed $\dot{\theta}_{t+\Delta t}$ is computed as follows

$$\dot{\theta}_{t+\Delta t} = (1-d)\dot{\theta}_t + \ddot{\theta}_t \Delta t \qquad (2)$$

where _d_ is a damping coefficient set to $5 \times 10^{-6}$ in our experiments. Pendulums with different lengths _L_ will exhibit different periods, with larger values of _L_ implying longer periods. Hence, a shorter pendulum manifests larger angular velocities $\dot{\theta}$ than a longer pendulum, making it more difficult to control.

## 2.2 Control task

In the experiment, participants were required to balance a pendulum vertically above its axis in a computer simulation and keep it upright for as long as possible. They controlled the pendulum using two keys of the computer's keyboard, corresponding to the controlled external mass accelerations $u = \pm 0.27$ ms$^{-2}$ respectively. To hold down a key exerts a continuous force until the key is released (i.e., the participant did not need to repeatedly press a key to exert a continuous force but just to hold it down). Note that both keys can act as brake or accelerator according to the direction of the pendulum.

The structure of the experiment was as follows. Prior to starting the experiment, an instructor demonstrated to the participants what is required to perform the task. Then, the participants performed 4 practice trials in which they learned how to control and balance the pendulum. The time allowed to balance the pendulum on each training trial was 3 minutes. In these practice trials, the length of the pendulum $L$ was varied to let participants experience tasks of different levels of difficulty: easier for longer pendulums (having smaller angular speed) and more difficult for shorter pendulums (having larger angular speed). Please note that we included a learning session to ensure that participants had sufficient familiarity and skill to address the task, not to study learning dynamics. Therefore, we did not analyze the data from the learning stage.

The main experiment consisted of 8 trials lasting 2 minutes each, in which participants were asked to balance pendulums of 8 different lengths $L$. The values of $L$ are: $L = 0.2, 0.3, 0.4, 0.5, 0.6, 0.7, 0.8$ and $0.9$ m, with $L = 0.2$ m being the most difficult control task and $L = 0.9$ m the easiest one. The experimental condition changed according to a within-participants design, so all participants were tested for all difficulty levels $L$ with the order of trials being randomized across them. By manipulating the pendulum length $L$ (and hence the speed at which the pendulum swings around its axis) we controlled the difficulty of the task. This allowed us to investigate the control strategies and the possible reduction of action and/or state variability with respect to the limits of human processing capacity.

More formally, the objective of each trial is to swing the pendulum from a starting configuration where it is still and downward ($\theta = 0$ rad and $\dot{\theta} = 0$ rad s$^{-1}$) in order to reach the upright configuration ($\theta = \pi$ rad). Then, participants were required to use the keys to keep the pendulum balanced in this upright position for as long as possible (i.e., $\theta \simeq 0$ rad and $\dot{\theta} \ll 1$ rad s$^{-1}$). We emphasise that in principle it is possible to design an artificial controller capable of balancing the pendulum of different lengths using the two provided mass accelerations $u$. However, in practice, when the task is addressed by humans, their information processing capacity is limited and therefore controlling shorter pendulums becomes increasingly hard.

Participants were informed that balancing the pendulums (especially the shorter ones) was challenging. They were told that failure to balance may be a possibility but were instructed to do their best in each case. Furthermore, participants were informed that the times for which the pendulum was balanced on its axis was measured.

## 3 Methods

Let us denote by $s_t \doteq (\theta_t, \dot{\theta}_t) \in \mathcal{S} \doteq [0, 2\pi) \times \mathbb{R}$ the pendulum's *state* at time $t$, being a two-dimensional tuple composed by the pendulum's angle $\theta$ and angular speed $\dot{\theta}$. In addition, let us denote with $a_t \in \mathcal{A} \doteq \{l, n, r\}$ the *action* that a participant selects at time $t$, being either to press the key that increases the pendulum's swing in the clockwise direction ("left" action

$a_t = l$), to press the key that increases the pendulum's swing in the anti-clockwise direction ("right" action $a_t = r$), or simply to do nothing ("no-action" $a_t = n$).

For each trial $i$, we collected the states $\vec{S}^i \doteq (s_0^i, s_{\Delta t}^i, s_{2\Delta t}^i, \ldots, s_{t_F}^i)$ visited by the pendulum and the actions $\vec{A}^i \doteq (a_0^i, a_{\Delta t}^i, a_{2\Delta t}^i, \ldots, a_{t_F}^i)$ chosen by the participant, where $\Delta t$ denotes the sample interval and $t_F$ the trial's final time. In the experiment we set $\Delta t \simeq 0.017$ s and $t_F = 120$ s, hence each trial $i$ is represented by the two time series $\vec{S}^i$ and $\vec{A}^i$, each one composed of $F = 7026$ samples.

To investigate the relationship between skilled control and the variability of actions and states, we analysed the aforementioned time series measuring the performance and estimating information-theoretic measures of variability for each participant in each experimental condition. To measure participants' performance we employed a mathematical framework commonly used in models of sequential decision-making and control [14, 15, 30]. We first defined the *reward* as a function of the state space $r : \mathcal{S} \rightarrow \{0, c\}$, which measure the immediate worth of a state with respect to the given balancing task. In the experiment we defined the reward as follows

$$r(s_t) = \begin{cases} 0 & \text{if} \quad (\theta_t \in (\pi - 0.27, \pi + 0.27) \text{ rad}) \ \ and \ \ (\mid \dot{\theta}_t \mid < 0.2 \text{ rad s}^{-1}) \\ c \in \mathbb{R}^- & \text{otherwise} \end{cases} \tag{3}$$

In other words, $r(s_t)$ penalises states that are not close to a balanced configuration, which are states where the pendulum is far from its upright position and the angular speed is far from zero. In the experiment we chose $c = -3 \times 10^{-3}$. Then, in order to evaluate an entire trial, we defined the *utility* of a trial $i$, denoted as $U^i$, as the sum of all rewards accumulated by the participant for all time steps of the trial. Hence, for trial $i$ we have

$$U^i \doteq \sum_{n=0}^{F} r(s_{n\Delta t}^i) \tag{4}$$

We quantify actions' and states' variability employing the information-theoretic notion of *entropy*, which is used to measure the uncertainty of a random variable [31]. Let us denote by $A^i \in \mathcal{A}$ the random variable representing the identically distributed actions chosen at every time step $t$ by the participant involved in the trial $i$. Let us also denote by $S^i \in \bar{\mathcal{S}}$ the random variable representing the identically distributed states visited at every time step $t$ by the pendulum during the trial $i$, where $\bar{\mathcal{S}}$ is a discretised version of $\mathcal{S}$ obtained through binning. Let us also assume that the time series $\vec{A}^i$ and $\vec{S}^i$ are realizations of sequential samplings of the random variables $A^i$ and $S^i$, respectively. Then, the entropy of $S^i$ is defined as

$$H(S^i) \doteq -\sum_{s^i \in \bar{\mathcal{S}}} P(s^i) \log P(s^i) \tag{5}$$

where we denoted by $P(s^i)$ the probability of the random variable $S^i$ being equal to the state $s^i$ (i.e., $Pr\{S^i = s^i\}$). The *conditional entropy* $H(A^i|S^i)$ quantifies the uncertainty left about the action $A^i$ once the state $S^i$ is known. It is defined as follows

$$H(A^i|S^i) \doteq -\sum_{s^i \in \bar{\mathcal{S}}} P(s^i) \sum_{a^i \in \mathcal{A}} P(a^i|s^i) \log P(a^i|s^i) \tag{6}$$

$H(S^i|A^i)$ is defined similarly. The reduction of uncertainty about $S^i$ once the variable $A^i$ is known (and vice versa) can be measured by the *mutual information* $I(S^i;A^i)$. This also quantifies in bits the amount of information that $S^i$ and $A^i$ have in common. The mutual information

between states and actions is defined as

$$I(S^i; A^i) \doteq \sum_{s^i \in \mathcal{S}} \sum_{a^i \in \mathcal{A}} P(s^i, a^i) \log \frac{P(s^i, a^i)}{P(s^i)P(a^i)} \tag{7}$$

The mutual information can also be rewritten as $I(S^i; A^i) = H(S^i) - H(S^i|A^i) = H(A^i) - H(A^i|S^i)$, i.e., it is symmetric.

## 4 Results

### 4.1 Information-theoretic data analysis

Here, we used information-theoretic measures to investigate the variability of the states $\vec{S}$ visited by the controlled system and the actions $\vec{A}$ chosen by the participants to control it. Specifically, we analysed the quantities $H(S^i)$, $H(A^i|S^i)$, $H(S^i|A^i)$ and $I(A^i; S^i)$ for all trials $i$. Interpreting the numerical values of information-theoretic quantities for continuous data requires some care, especially when considering differential entropy. The latter conceptually does not depend on a arbitrarily chosen bin size, but is, strictly spoken, not directly interpretable as entropy; amongst other, it might become negative or diverge (see [32], sec. 8–2, p. 269). More specifically, differential entropy is essentially the limit of entropy obtained by binning when the binning becomes arbitrarily fine, subtracting a logarithmic term correcting for the fineness of the binning; the differential entropy is therefore a "renormalized" version of the binned entropy in the limit of arbitrarily fine binning. As a consequence, differential entropies have no simple interpretation, only their differences do. Another alternative could have been to employ entropy rates. However, given the available data set, a problem in this regard is that we do not have an ensemble of trajectories that would have allowed us to average across several identical processes. Here, thus, we instead consider entropies which we obtain from continuous distributions by first binning them. These quantities do have a direct interpretation as entropies, but their value is shifted by an arbitrary offset which depends only on the choice of binning. Thus, it is possible to consistently compare binned entropies relative to each other, as long as the binning used to compute them is the same. This is the approach we use here. To estimate the above quantities from the time series data collected during the experiment, we adopted the numerical estimator MIToolbox [33], which employs conditional likelihood maximization to compute the entropy. To use this estimator for discrete random variables and estimate the quantities of trial $i$, we sampled the state time series of trial $i$ at a fix rate. We then discretized the sampled time series binning the angle interval $[0, 2\pi)$ rad into 1000 bins of equal size and the angular speed interval $[-10, 10]$ rad s$^{-1}$ into 200 bins of equal size.

**4.1.1 Direct relation between utility and action entropy in skilled action control.** The first question we addressed is whether, in keeping with classical theories of action routinization, skilled control is indexed by low levels of action entropy, given the states where the actions are executed, i.e., $H(A^i|S^i)$. For this, we investigated how the utility $U^i$ (and hence participants' skill in the pendulum balancing tasks) varies as a function of action entropy. We found a direct relation between the utility achieved by participants and the entropy of their action distributions $H(A^i|S^i)$ (see Fig 2A). Significant positive Spearman and Pearson correlations were observed between $H(A^i|S^i)$ and $U^i$ ($\rho = 0.944$, $p < 0.01$; $r = 0.967$, $p < 0.01$) for the whole set of trials $i$, ignoring $L$, which shows a strong monotonic relationship between the two variables (i.e., when the entropy $H(A^i|S^i)$ increases the utility $U^i$ tends to increase). For correlation analyses, we used IBM SPSS Statistics (Version 25.0). In Fig 2A, each data point is coloured according to the difficulty level of the corresponding trial (i.e., pendulum length $L$). As expected, utilities $U^i$ are higher for (longer) pendulums that are easier to control than for

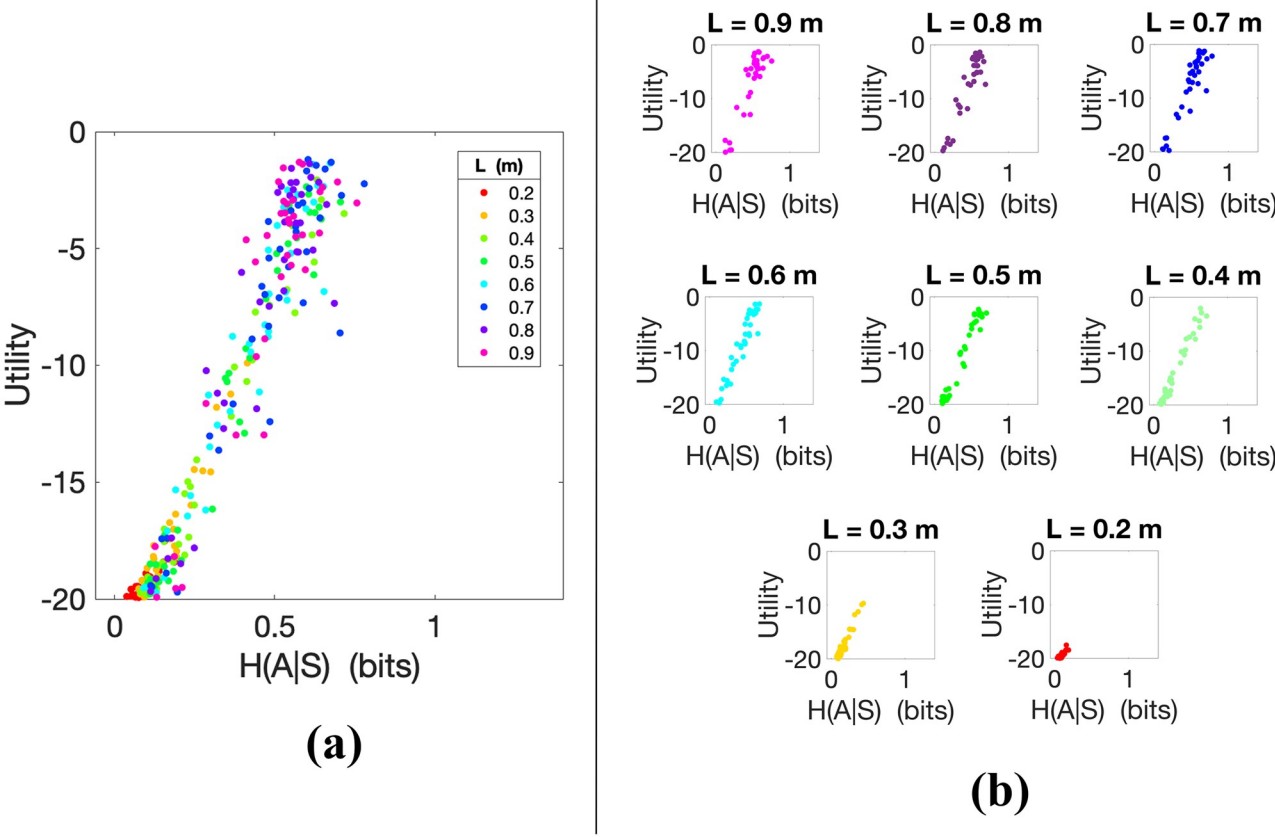

**Fig 2.** **(a)** Utility $U^i$ of trials $i$ during the balancing task plotted as a function of $H(A^i|S^i)$ for all participants and pendulum lengths $L$. Larger values of utility $U^i$ correspond to better performance in balancing the pendulum. Spearman correlation $\rho = 0.944$; Pearson correlation $r = 0.967$. In **(b)** the trials are separated by pendulum length in different subplots.

(shorter) pendulums that are more difficult to control. Participants who controlled the pendulums better had a larger variability in their action distribution, represented by a larger conditional action entropy $H(A^i|S^i)$, as visible in the figure. To test whether these results are general across pendulum lengths, in Fig 2B we show the action entropy $H(A^i|S^i)$ of the eight different pendulum lengths $L$ separately, with one subplot for each pendulum length. The positive Spearman correlation between utility and action entropy is consistent across all values of $L$, as shown in Table 1.

Finally, to further investigate the relations between the pendulum length $L$ and action entropy $H(A|S)$, in Fig 3 we plot the action entropy $H(A|S)$ averaged across all the trials with the same pendulum length $L$. The figure shows a directly proportional relationship between average action entropy and pendulum length. Assuming that participants are generally more proficient in controlling longer pendulums, this plot confirms that higher skill levels are indexed by larger action entropy.

Taken together, these results run against the idea that skilled control is indexed by low levels of action entropy—and indicate instead that they are indexed by high levels of action entropy.

**4.1.2 Inverse relation between utility and state entropy in skilled action control.** The second question we addressed is whether, in keeping with perceptual control theory, skilled control is indexed by low levels of state entropy $H(S^i)$ and/or state action-conditioned entropy $H(S'^i|A^i)$. For this, we considered how utility $U^i$ varies as a function of these two entropies. By

**Table 1. Action and state entropies' Spearman coefficients ($\rho_{H(A^i|S^i),U^i}$ and $\rho_{H(S^i),U^i}$, respectively) for all pendulum lengths $L$.**

| $L$ | $\rho_{H(A^i|S^i),U^i}$ | $\rho_{H(S^i),U^i}$ |
|---|---|---|
| 0.2 m | 0.770 ($p < 0.01$) | -0.747 ($p < 0.01$) |
| 0.3 m | 0.902 ($p < 0.01$) | -0.951 ($p < 0.01$) |
| 0.4 m | 0.979 ($p < 0.01$) | -0.992 ($p < 0.01$) |
| 0.5 m | 0.939 ($p < 0.01$) | -0.978 ($p < 0.01$) |
| 0.6 m | 0.953 ($p < 0.01$) | -0.971 ($p < 0.01$) |
| 0.7 m | 0.844 ($p < 0.01$) | -0.974 ($p < 0.01$) |
| 0.8 m | 0.751 ($p < 0.01$) | -0.976 ($p < 0.01$) |
| 0.9 m | 0.719 ($p < 0.01$) | -0.951 ($p < 0.01$) |
| all Lengths | 0.944 ($p < 0.01$) | -0.982 ($p < 0.01$) |

$S'^i$ we denote the random variable indicating the state visited by the pendulum 270 ms after an action $A^i$ has been selected. This interval approximates the estimated time that elapses to execute a motor control (e.g., for a saccade is between 250–300 ms; see [34]). The quantity $H(S'^i|A^i)$ can be interpreted as the amount of uncertainty left in the joint distribution $P(A^i, S'^i)$ about the state $S'^i$ once the uncertainty about the action $A^i$ has been removed. A low $H(S'^i|A)$ means that to know the chosen action $A^i$ implies small uncertainty and variability about the resulting state $S'^i$.

Figs 4A and 5A show an inverse relation between utility and both the entropy $H(S^i)$ of states and the entropy $H(S'^i|A^i)$ of future states given participants' actions. As in the previous analysis, utilities $U^i$ are higher for pendulums that are easier to control than for pendulums that are more difficult to control. The Spearman coefficients of the utility $U^i$ with the entropies $H(S^i)$

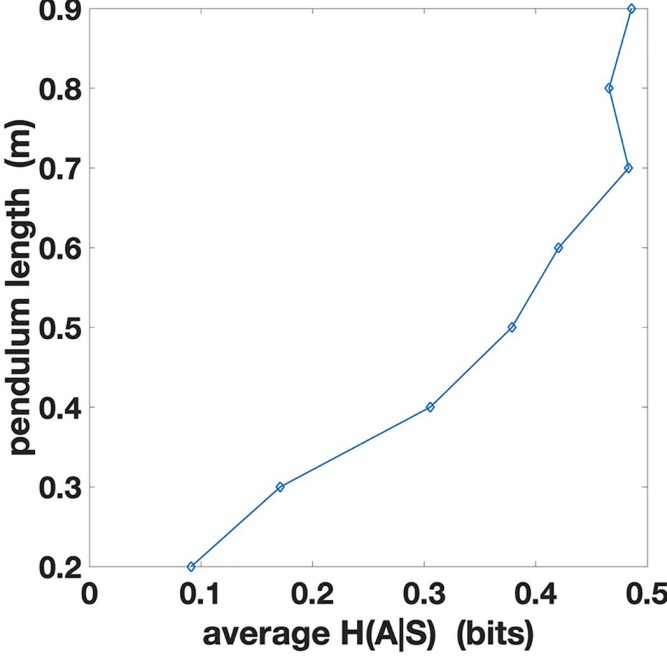

**Fig 3. Action entropy $H(A|S)$ averaged over trials with the same pendulum length $L$.** The abscissa represents the action entropy and the ordinate represents the pendulum length to allow direct comparison with Fig 2.

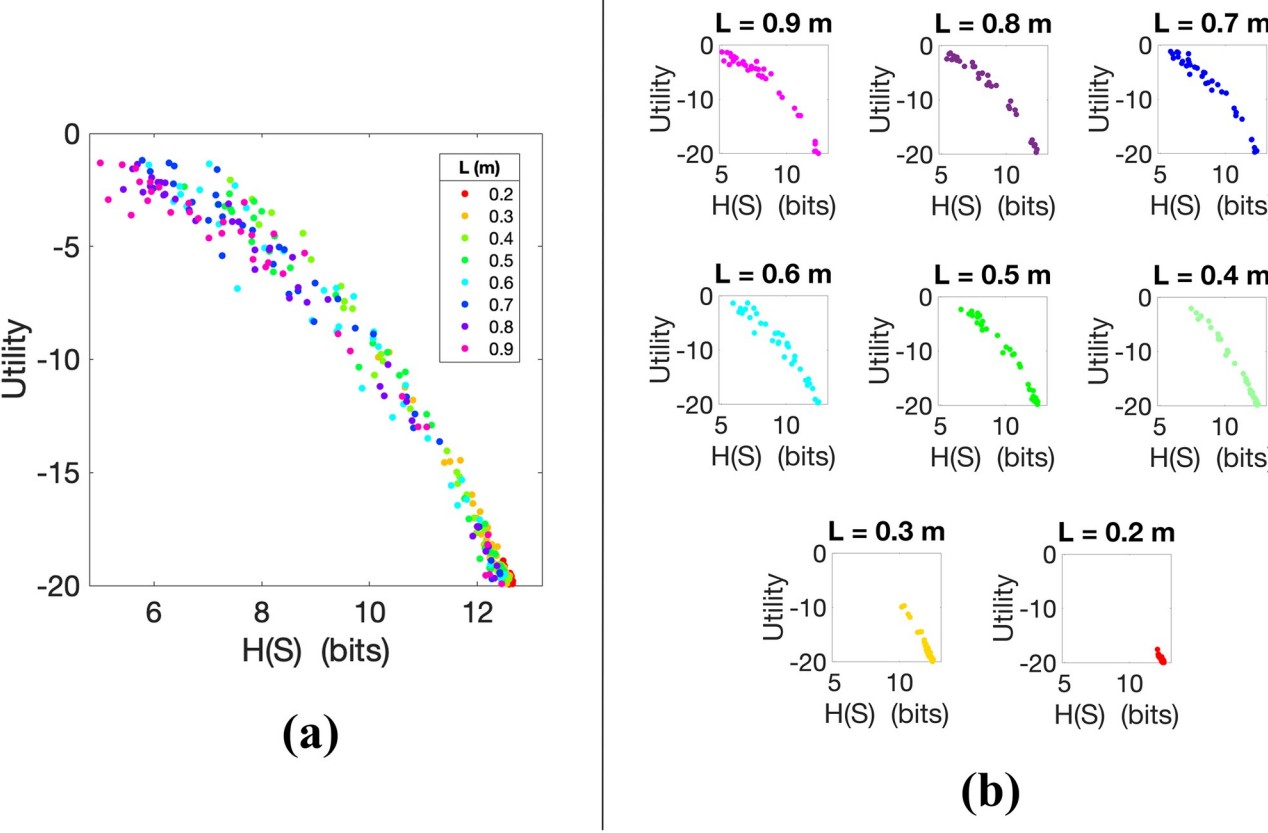

**Fig 4.** **(a)** Utility $U^i$ of trials $i$ during the balancing task plotted as a function of $H(S^i)$ for all participants and pendulum lengths $L$. Spearman correlation $\rho = -0.982$. In **(b)** the trials are separated by pendulum length in different subplots.

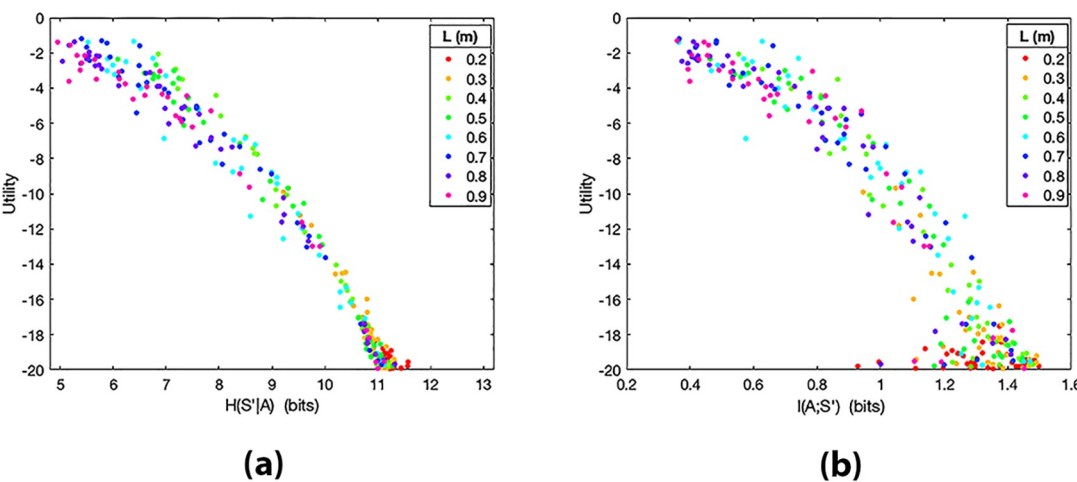

**Fig 5. Utility $U^i$ of trials $i$ during the balancing task plotted as a function of $H(S'^i|A^i)$ in (a) and of $I(A^i;S'^i)$ in (b), for all participants and pendulum lengths $L$. (a)** Spearman correlation $\rho = -0.979$. **(b)** $\rho = -0.907$.

($\rho = -0.982$, $p < 0.01$) and $H(S'^i|A^i)$ ($\rho = -0.979$, $p < 0.01$) were computed for all trials $i$ and independently from $L$, indicating a strong correlation in the negative direction (i.e., when the state entropies decrease the utility tends to increase).

As Fig 4B shows, the monotonic relationship between state entropy $H(S^i)$ and utility is consistent across all pendulum lengths $L$. Interestingly, the data points of different pendulum lengths lie approximately on the same curve and can be nicely matched up, as shown in Fig 4A (i.e., more capable subjects balancing short pendulums, which are more difficult to control, are quite comparable to less skilful subjects balancing longer pendulums, which are easier to control). The possible origins of this alignment will be discussed in Section 5. The negative Spearman correlation between utility and state entropy is also consistent across all values of $L$, as shown in Table 1. Taken together, these results lend support to the hypothesis that skilled control is indexed by low *state* entropy, irrespective of the values of $L$. They are also compatible with the idea that expert participants perform actions that decrease the entropy of their future states.

To further investigate the relations between pendulum length $L$ and state entropy $H(S)$, in Fig 6 we plot the state entropy $H(S)$ averaged across all the trials with the same pendulum length $L$. The figure shows that average $H(S)$ is lower for longer pendulums, which are easier to control, and greater for shorter pendulums, which are more challenging to control. Hence, although different values of $L$ lead to similar state entropy trends, these induce different regimes with respect to state information processing.

The two entropies $H(S^i)$ and $H(S'^i|A^i)$ in Figs 4A and 5A have a similar trend, but the latter is slightly smaller on average (their difference, averaged for all trials, is 1.03 bits and close to 1.5 bits for less skilled participants). This shows that most of the information contained in $H(S'^i|A^i)$ comes from the information contained in $H(S^i)$ but, compared to the latter, is reduced by an average difference of about 1 bit. So, although most of the information content is carried by state information, some information is absorbed (i.e., can be predicted) by the

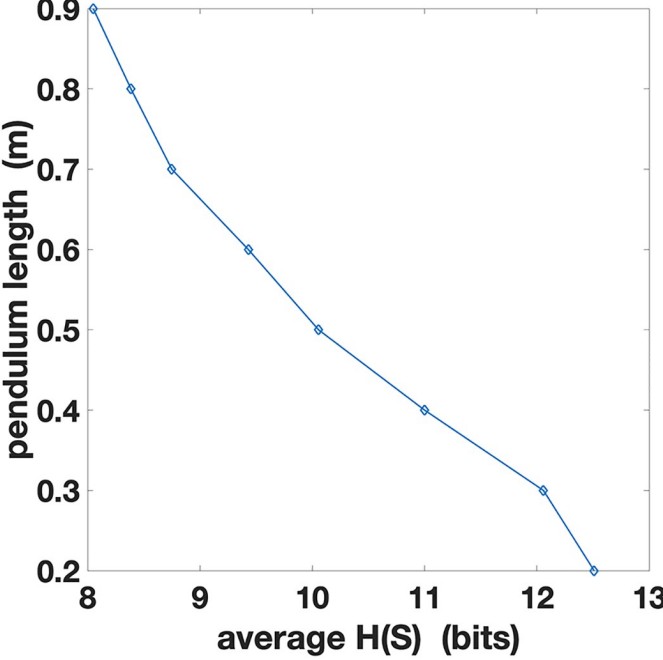

**Fig 6. State entropy $H(S)$ averaged across trials with the same pendulum length $L$.** The abscissa represents the state entropy and the ordinate represents the pendulum length to allow direct comparison with Fig 4.

action selected by participants. This represents the amount of information the actions $A^i$ provide about the resulting states $S'^i$ independently from the previous states $S^i$. Note that this information is not far from $\log_2(3)$, i.e., the information that distinguishes between the three regions that the action is pushing the pendulum towards by choosing $r$, $l$ or $n$. Since $S'^i$ is essentially $S^i$ shifted in time, we can assume that $H(S^i)$ is equal to $H(S'^i)$. Hence, the aforementioned difference becomes the mutual information $I(A^i;S'^i) = H(S'^i) - H(S'^i|A^i)$ (see Fig 5B), which removes from the information contained in $S'^i$ the information contained in $S'^i$ knowing already $A^i$ [35–39]. Note that the results involving the mutual information $I(A^i;S'^i)$ are consistent with those obtained with the Kraskov-Stögbauer-Grassberger (KSG) method for estimating the mutual information for time series with continuous data [40], which was used in addition to [33] to verify our findings. These results are not presented here because they do not provide additional insights.

The mutual information $I(A^i;S'^i)$ is reported here for consistency although in character it is just reflecting the entropies. The fact that $I(A^i;S'^i)$ is basically just the information injected by the action shows that the environment does not absorb the action entropy, but rather determines the subsequent behaviour. In other words, the actions chosen by the participants have a significant effect on the resulting behaviour of the pendulum. The Spearman correlation of the mutual information $I(A^i;S'^i)$ with the utility $U^i$ for all $i$ ($\rho = -0.907$, $p < 0.01$) shows the similar strong negative trend found for the state entropies. The narrow and neat character of the curves reported in Figs 4 and 5 is a strong indicator that their trends are general across participants rather than reflecting the particular decisions of a subset of them. The only case where the data points are more spread is the wide "foot" of the mutual information curve, which shows that when the pendulum becomes very hard to control, the relationship between information and utility becomes fuzzier.

Similarly to our findings, in Lupu et al. [28, 41] the average information-transmission rate of humans balancing an inverted pendulum is shown to be inversely related with the pendulum's length. The latter is used by the authors to parameterize the system's stability and the time delays experienced by the human participants. In their work, the mutual information rate of the control systems is considered and is estimated using the Kolmogorov-Sinai entropy, whereas in our study we estimated the Shannon mutual information of states and actions directly from data.

To further analyze the relationship between utilities and state entropy, we plotted the support of $S^i$ over the phase spaces of two pendulums: the former longer and easier to control ($L = 0.9$ m, see Fig 7A and 7B) and the latter shorter and more difficult to control ($L = 0.2$ m, see Fig 7C and 7D). Our results indicate that the support of $S^i$ is smaller when the pendulum is controlled successfully – as it is the case for most participants with a long pendulum (see Fig 7A and 7B for two examples). On the contrary, the support of $S^i$ is wider when the pendulum is not controlled properly—as it is the case for most participants with a short pendulum (see for instance Fig 7C and 7D). This analysis further supports the idea that a hallmark of skilled action control consists in maintaining a low entropy of state distributions across the whole task, providing a visual illustration of the notion of a "narrow corridor" in the phase space of the task mentioned in the Introduction.

## 4.2 Analysis of time spent in the balancing region

In this section, we ask whether the relations between the utility $U^i$ and the entropies $H(A^i|S^i)$, $H(S^i)$ and $H(S'^i|A^i)$ reported in Section 4.1 depend on the time participants spend in the balancing area and on the specific strategies they adopt in that area. Since the pendulum is an inherently unstable system, balancing the pole in the inverted position requires moving it

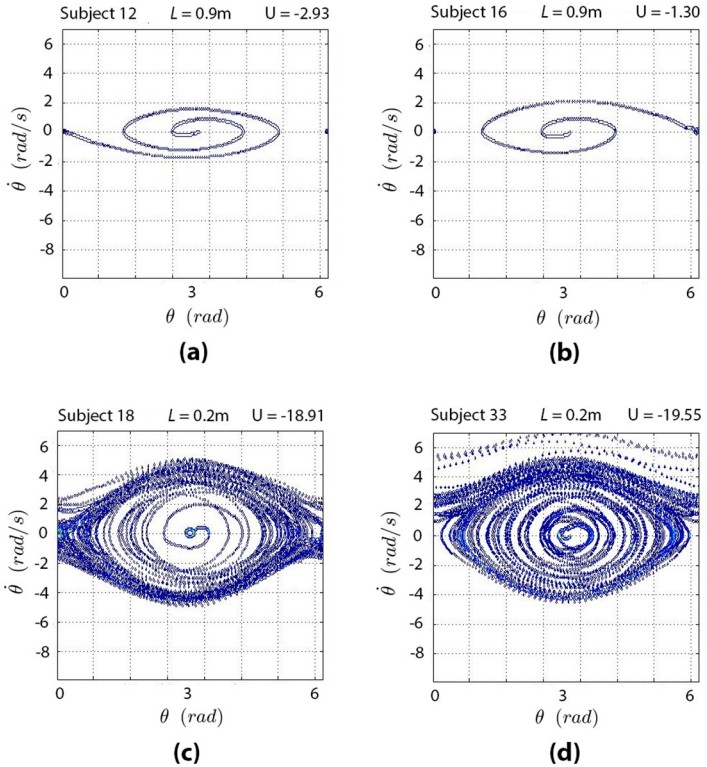

**Fig 7. Support of $S^i$ within the phase space of four representative trials: Two participants swinging the longest pendulum ((a) and (b)) and other two participants swinging the shortest pendulum ((c) and (d)).** Trials start at the center of the phase space.

continuously. Therefore, the fact that more skilled participants spend more time in the balancing region, alternating the actions $l$ and $r$ repeatedly to keep the pendulum upright, might explain the direct relation between $U^i$ and $H(A^i|S^i)$. Furthermore, when the pendulum is balanced in its upright position, it visits a relatively little area of its phase space and consequently has a small support of the state distribution. Hence, the fact that more skilled participants consistently spend more time in the balancing region might explain the inverse relation between $U^i$ and the entropies $H(S^i)$ and $H(S'^i|A^i)$, because the support's size of a distribution is one of the factors that impact the magnitude of its entropy. To validate these hypotheses, we investigated the different strategies that participants adopt when they are in different phases of the task: namely, when they were within or outside the balancing region.

First, as a sanity check, we tested whether participants spend less time in the balancing region when the pendulum is more difficult to control. The analysis shown in Fig 8 confirms this prediction. Furthermore, the analysis of button presses within or outside the balancing regions indicates that participants do more key presses in the balancing region (Fig 9)—as several micro-adjustments may be necessary to keep the pendulum in the correct position – increasing action variability for skilled participants.

A more detailed analysis of the actions performed by participants in the balancing region is shown in Fig 10. The figure shows the distance travelled by the pendulum when a key is down, with blue segments representing actions that push the pendulum in the clockwise direction and red segments representing actions that push the pendulum in the anti-clockwise direction. Subsequent key presses are reported one below another, starting from the top of the plot. The three plots of Fig 10 permit appreciating the differences between the key presses required to

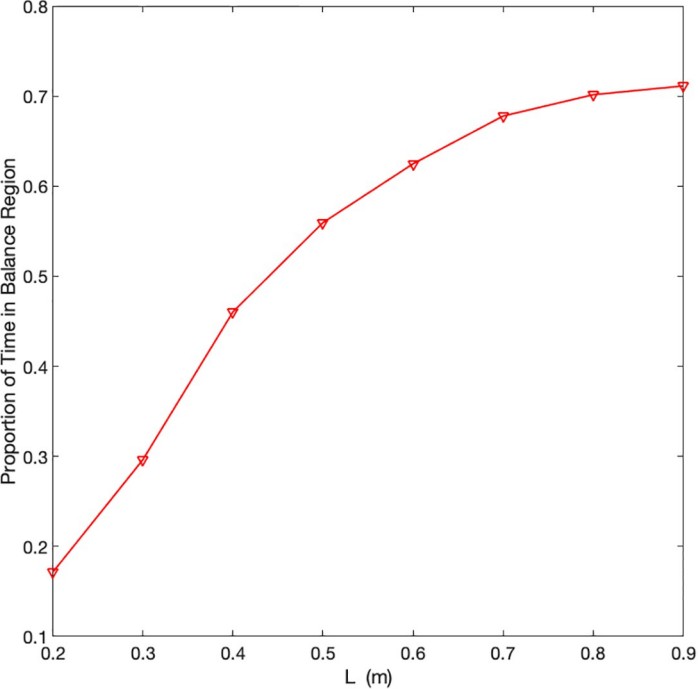

**Fig 8. Proportion of the time spent by participants in the balancing region (± 0.27 rad from the vertical axis) for each pendulum length *L*.**

balance a pendulum that is easier to control (*L* = 0.9 m, see Fig 10A) versus a pendulum that is more difficult to control (*L* = 0.5 m, see Fig 10B and 10C). Furthermore, the figures show the differences between selected participants with higher skill (Fig 10A and 10B) and lower skill levels (Fig 10C).

In general, controlling successfully the pendulum in the balancing region (keeping it upright) requires large variability in the choice and timing of actions (Fig 10A and 10B). This phenomenon is especially present in the case of the more skilled participant who controls the more challenging pendulum (Fig 10B). This finding generalises to the typical behaviour of more skilled participants, who spend more time in the balancing region (also during difficult tasks) and manifest a large variability in their actions, compared to less skilled participants who spend less time in the balancing region. However, less skilled participants perform fewer key presses in the balancing region (see Fig 10C), manifesting a smaller variability in their actions. This analysis shows the strong task-dependence of the relations between the utility $U^i$ and the entropies $H(A^i|S^i)$, $H(S^i)$ and $H(S'^i|A^i)$, see also Section 5.

### 4.3 The role of noise in the utility-information curves: A computational study

In this section, we report the results of a series of simulations that we performed to identify the source of state and action variability observed in the data. Since the dynamics of the pendulum controlled by the participants is deterministic, all the variability measured through our information-theoretic analysis must come from the actions chosen by the participants. The results of our numerical study show that the low state variability that we observed in skilled participants is a direct consequence of the dynamics of the controlled pendulum, which induces some mandatory trajectories in order to be balanced. Furthermore, the direct relation between

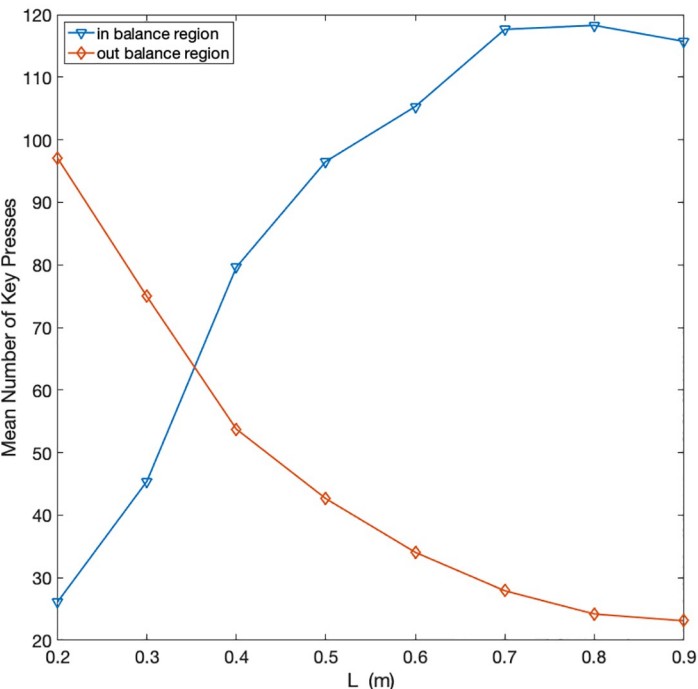

**Fig 9. Key press activity within and outside balance region (± 0.27 rad from the vertical axis) for each pendulum length $L$.**

action variability and participants' performance may be explained by both the presence of noise in the participants' observation of the pendulum's state and by their utilization of a stochastic model of the pendulum's dynamics. Concerning this latter point, it is important to remind that the most skilled participants spent more time in the balancing region, where the frequency of key presses is large. In this regard, a deterministic controller would always choose identical actions in the same states, whereas a non-deterministic controller may choose different actions in the same states, increasing the action variability. Hence, the large action entropy of skilled participants might depend on the fact that they adopt a stochastic decision-making process.

To this aim, we employed an artificial controller that solves the same pendulum balancing task addressed by the human participants in the reported experiment (Section 2.2). To simulate their behavior, we used a computational model based on Markov decision processes (MDP) [14, 15]. We used a discrete MDP defined by the tuple $(\hat{\mathcal{S}}, \mathcal{A}, T, r)$, which is composed as follows: $\hat{\mathcal{S}}$ is a discrete version of the state space $\mathcal{S}$ obtained through uniform binning, where we used 1000 bins for the angle $\theta$ and 1000 bins for the angular speed $\dot{\theta}$; the action space $\mathcal{A}$ and the reward function $r$ are defined as in Section 3; $T(\hat{s}'|\hat{s}, a)$ is the transition probability distribution, with $T : \hat{\mathcal{S}} \times \mathcal{A} \times \hat{\mathcal{S}} \to [0, 1]$. In our MDP-based controller, $T$ is characterized by the random disturbance $\delta \doteq \delta_\theta \times \delta_{\dot{\theta}}$, where $\delta_\theta \simeq \mathcal{N}(0, \sigma_\theta^T)$ and $\delta_{\dot{\theta}} \simeq \mathcal{N}(0, \sigma_{\dot{\theta}}^T)$ are discrete approximations (via binomial distribution [42]; [43] see sec. 4.5, p. 105) of normally distributed random variables added to the difference equations of the pendulum's angle and angular speed respectively (via Eqs (1) and (2)). Furthermore, we included in the controller a simple sensor model intended to mimic the perceptual disturbances that participants may have experienced while controlling the pendulum. This is characterized by the observation function $O : \hat{\mathcal{S}} \to \Omega$, defined as $O(\hat{s}) = \hat{s} + \eta$, with observation space $\Omega = \hat{\mathcal{S}}$ and observation

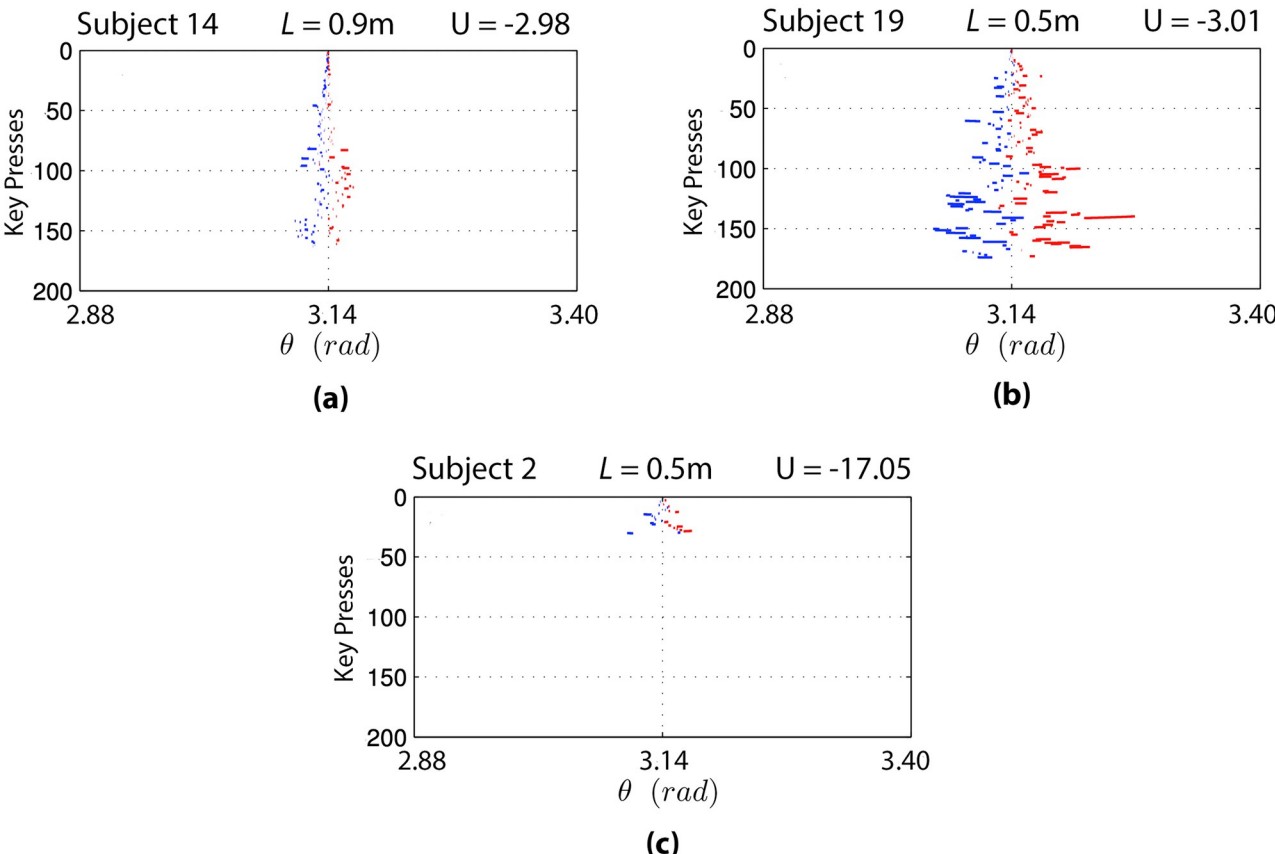

**Fig 10. Distribution of key presses in the balancing region for three trials and distance traveled by the pendulum during the key presses.** Blue segments represent actions that push the pendulum in the clockwise direction, red segments indicate actions that steer the pendulum towards the anticlockwise direction. Key presses ("balancing episodes") are reported subsequently from the top to the bottom of the plot. In **(a)** and **(b)** the key presses of two skilled participants are reported, where in **(c)** the key presses of a less skilled participant is shown.

random disturbance $\eta \doteq \eta_\theta \times \eta_{\dot\theta}$. Also the observation disturbances $\eta_\theta \simeq \mathcal{N}(0, \sigma_\theta^O)$ and $\eta_{\dot\theta} \simeq \mathcal{N}(0, \sigma_{\dot\theta}^O)$ are discrete binomial approximations of normal random variables.

To enable the numerical study of the relationship between state and action variabilities and the participants' performance observed in our experiments, the skill level of the employed controller needs to be parameterized. In this regard, we chose as baseline for the highest skill level the optimal policy $\pi^*$, where we denote a behavioral policy with $\pi : \hat{\mathcal{S}} \to \mathcal{A}$, which is a state-action map that tells the controller what action to take in every state. For every state $\hat{s}$ and time $t$, the optimal policy is defined as $\pi^*(\hat{s}) \doteq \mathrm{argmax}_\pi \mathbb{E}_\pi[\sum_{k=0}^F r(\hat{s}_{t+k+1})|\hat{s}_t = \hat{s}]$, where the expected value is taken over all the trajectories generated by using the policy $\pi$. Given that $\pi^*$ represents the controller with maximum skill level, this can be progressively degraded using an epsilon-greedy strategy [15]. The epsilon-greedy policy is defined as

$$
\pi^{\epsilon_n}(\hat{s}) \doteq
\begin{cases}
\pi^*(\hat{s}) & \text{with probability } 1 - \epsilon_n \\
\mathcal{U}\{l, n, r\} & \text{with probability } \epsilon_n
\end{cases}
\tag{8}
$$

where we denoted with $\mathcal{U}$ the uniform distribution and with $\epsilon_n$ the probability of having the controller choosing a random action. The epsilon-greedy probability is defined as $\epsilon_n \doteq \frac{n}{N}$, with

$n = 0, 1, \ldots, N$ indicating the controller skill level. Hence, the highest skill level $n = 0$ represents the optimal controller, the lowest skill level $n = N$ corresponds to a completely random controller and $0 < n < N$ denotes controllers with intermediate skill levels. To compute the baseline optimal policies $\pi^*$ we used the value iteration algorithm [15] combined with a k-d tree data structure [44], which is used to efficiently handle the binned state space. The role of the proposed sensor model is to add noise to the decision-making process, meaning that when the MDP is in state $\hat{s}$ the controller selects its action according to $\pi^{\epsilon_n}(O(\hat{s}))$ (no partially observable Markov decision processes (POMDP) computation is used [45]).

We conducted a series of numerical experiments having artificial controllers of $N = 26$ different skill levels addressing the same pendulum balancing task described in Section 2.2. In this regard, the same pendulum simulator was used to run 26 simulations for each of the 8 pendulum lengths. Each simulation $j = 1, 2, \ldots, 208 = 26 \times 8$ lasted the same number of time steps used for the human trials, yielding time series $\vec{S}^j_\pi$ and $\vec{A}^j_\pi$ of the same size $F$. The data analysis presented in Section 4.1.1 and 4.1.2 was carried out for the time series generated by the artificial controllers of all simulations $j$, including the computation of the utilities $U^j$ and the information-theoretic quantities $H(A^j|S^j)$, $H(S^j)$, $H(S^j|A^j)$ and $I(A^j;S^j)$. In addition, to investigate the impact that the model and observation noise might have in determining the information-utility curves that we reported in the previous sections, here we use the artificial controllers to show how action and state variability relate with utility for disturbances $\eta$ and $\delta$ of different magnitude. In the following, we present scatter plots of the utility as a function of the quantities $H(A^j|S^j)$ and $H(S^j)$ for different values of $\sigma^T_\theta$, $\sigma^T_{\dot\theta}$, $\sigma^O_\theta$ and $\sigma^O_{\dot\theta}$. We do not report the results regarding $H(A^j|S^j)$ and $I(A^j;S^j)$ because these do not provide further relevant insights. Similarly to the previous sections, each data point within the scatter plots represents a different simulation $j$ and data points are colored according to pendulum length $L$.

**4.3.1 Deterministic artificial controllers.** In Fig 11, we show the entropies $H(A^j|S^j)$ and $H(S^j)$ for controllers that are not affected by any disturbances. While Fig 11B resembles the inverse relationship between utility and state entropy found analyzing the human data (controllers' Spearman correlation $\rho = -0.960$, $p < 0.01$; participants' $\rho = -0.982$), the action entropy plot (Fig 11A) shows a different trend from the direct relationship observed in the experiment (controllers' $\rho = 0.67$, $p < 0.01$; participants' $\rho = 0.944$). In fact, we can see that for large utility values the action entropy is decreasing rather than increasing. Since to achieve a large utility the controller needs to keep the pendulum balanced in its upright position, alternating repeatedly $l$ and $r$ actions, this can be explained by the fact that, in absence of noise, the same actions are chosen in the same states, decreasing the action entropy. Another difference with the human data analysis worth noting is the relation between utility and pendulum length. Indeed, due to human limited processing capacity, participants find longer pendulum easier to control and, in doing so, when compared with short pendulums, they obtain higher utility. On the contrary, in Fig 11 (and the following plots) we see that longer pendulums have lower utility than shorter ones. This due to the fact that we decided to not include any limitation in terms of processing capacity in the employed controllers, because the focus of our study is the relationship between utility and variability of state and action. Since longer pendulums are slower, and the employed reward function penalizes states that are outside the balancing area, the artificial agents achieve less utility when controlling them as they take more time to reach the goal state. Furthermore, we will see that slower pendulum are more sensitive to angular speed errors. A possible way to address this discrepancy would be assigning different levels of noise according to the length or speed of the pendulums. But, since we do not have a concrete hypothesis regarding how to distribute the noise levels according to pendulum length or speed, we leave this aspect to future work.

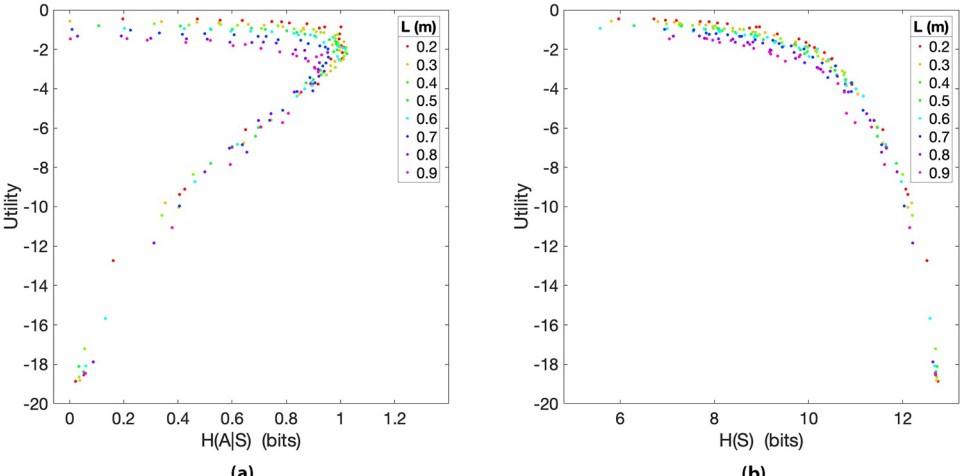

**Fig 11. Utility $U^j$ of simulations $j$ plotted as a function of $H(A^j|S^j)$ ((a)) and $H(S^j)$ ((b)) for all artificial controllers and pendulum lengths $L$.** These can be compared to Figs 2A and 4A respectively. No observation and transition noise is employed here (i.e., $\sigma_\theta^O = \sigma_{\dot\theta}^O = \sigma_\theta^T = \sigma_{\dot\theta}^T = 0$). **(a)** Spearman correlation $\rho = 0.67$ ($p = 0.334$, not significant, due to the marked non-monotonicity of the data). **(b)** $\rho = -0.96$ ($p < 0.01$).

**4.3.2 Artificial controllers with observation noise.**  We hypothesize that the introduction of a disturbance will lead to an increase of action variability for large utility values, therefore we made the cognitively plausible choice of introducing the disturbance $\eta$ in the controllers' observation of the pendulum state. Since the policies $\pi^{\epsilon_n}$ are computed over states rather than observations, controllers choose actions via $\pi^{\epsilon_n}(O(\hat{s}))$ for observations $O(\hat{s})$ that may be different from the actual states of the pendulum $\hat{s}$. Consequently, the "entanglement" between actions and states is reduced and the entropy $H(A^j|S^j)$ is increased.

In Fig 12, scatter plots of $H(A^j|S^j)$ and $H(S^j)$ for increasing magnitudes of the observation disturbances $\eta$ are reported. In Fig 12A, we can observe that with $\sigma_\theta^O = 0.01$ and $\sigma_{\dot\theta}^O = 0.02$ the action entropy for large utility values is increased. Furthermore, in Fig 12C, by increasing the observation standard deviations to $\sigma_\theta^O = 0.1$ and $\sigma_{\dot\theta}^O = 0.2$, we obtain a highly direct relationship between utility and action entropy with Spearman correlation $\rho = 0.908$ ($p < 0.01$) and Pearson correlation $r = 0.980$ ($p < 0.01$), which are close to the correlations found analyzing the human data ($\rho = 0.944$; $r = 0.967$). Another consequence of including observation disturbances is to increase the state entropy $H(S^j)$ for simulations that led to large utility (the minimum of $H(S^j)$ is 7.95 bits in Fig 12D, whereas it is 5 bits in Fig 4A), because the presence of noise introduces variability in the system. Finally, by increasing further the standard deviations to $\sigma_\theta^O = 1$ and $\sigma_{\dot\theta}^O = 2$, we can see that the larger noise significantly impacts the performance of the controller, causing the overall achieved utility to decrease and some of the longer pendulums to not be balanced at all.

**4.3.3 Artificial controllers with transition and observation noise.**  In the previous section, we have seen that to introduce observation disturbances has the effect of increasing both $H(A^j|S^j)$ and $H(S^j)$ for large utility values. Comparing the state entropy's scatter plot of the artificial controller with observation noise (Fig 12D) with the one of the human data (Fig 4A), we notice that the participants are more efficient in reducing the state variability (i.e., the minimum of their $H(S^j)$ is lower). In this regard, our numerical study has shown that when agents model the pendulum dynamics with slightly stochastic transitions, the observation random disturbances are regularized, increasing a little the average performance achieved by the controllers and reducing $H(S^j)$ when the utility is large. This can be explained by the fact that

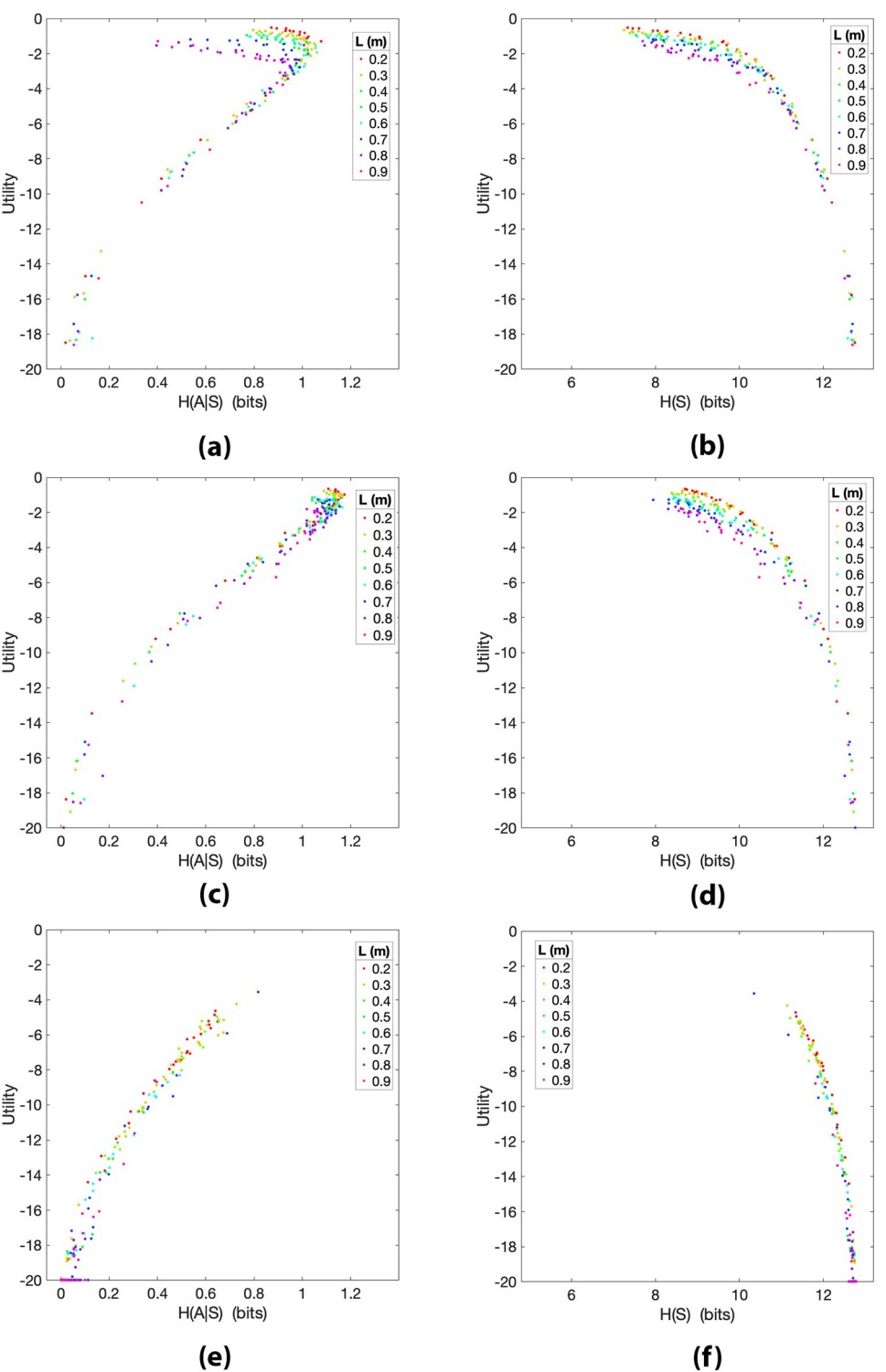

**Fig 12. Utility $U^j$ of simulations $j$ plotted as a function of $H(A^j|S^j)$ ((a,c,e)) and $H(S^j)$ ((b,d,f)) for all pendulum lengths $L$ and artificial controllers with different observation disturbances $\eta$ (here $\sigma_\theta^T = \sigma_{\dot\theta}^T = 0$). (a)** Observation standard deviations $\sigma_\theta^O = 0.01$, $\sigma_{\dot\theta}^O = 0.02$; Spearman correlation $\rho = 0.474$ ($p < 0.01$). **(b)** $\sigma_\theta^O = 0.01$, $\sigma_{\dot\theta}^O = 0.02$; $\rho = -0.951$ ($p < 0.01$). **(c)** $\sigma_\theta^O = 0.1$, $\sigma_{\dot\theta}^O = 0.2$; $\rho = 0.908$ ($p < 0.01$), Pearson correlation $r = 0.980$ ($p < 0.01$). **(d)** $\sigma_\theta^O = 0.1$, $\sigma_{\dot\theta}^O = 0.2$; $\rho = -0.908$ ($p < 0.01$). **(e)** $\sigma_\theta^O = 1$, $\sigma_{\dot\theta}^O = 2$; $\rho = 0.954$ ($p < 0.01$). **(f)** $\sigma_\theta^O = 1$, $\sigma_{\dot\theta}^O = 2$; $\rho = -0.954$ ($p < 0.01$).

when an observation $o = O(\hat{s})$ is different from the actual state $\hat{s}$, the controller acts according to $\pi^{\epsilon_n}(o)$ with the false belief that the next state will be $o'$ rather than the actual successor state $\hat{s}'$. Hence, when $\pi^{\epsilon_n}$ is computed considering a non deterministic transition function with $T(\hat{s}'|o, \pi^{\epsilon_n}(o)) > 0$, the error that the agent would have committed assuming that the successor is $o'$ may be reduced due to optimal policy maximization of *average* cumulated reward. Note that, although this simple non-optimal approach was sufficient to reduce the state entropy for high utility values, in artificial intelligence the optimal policy of an agent with noisy sensors can be computed using POMDP [45].

In Fig 13A and 13B, we report $H(A^j|S^j)$ and $H(S^j)$ for $\sigma_{\hat{\theta}}^O \simeq 0.01$ and $\sigma_{\hat{\theta}}^O \simeq 0.07$. With these observation disturbances $\eta$, when the utility approaches its maximum, the action entropy slightly decreases and the state entropy reaches a minimum of 8 bits. In Fig 13C and 13D, we kept the observation disturbances $\eta$ of the previous plots and introduced transition disturbances $\delta$ with standard deviations $\sigma_{\theta}^T \simeq 0.007$ and $\sigma_{\hat{\theta}}^T \simeq 0.014$. In Fig 13C, the combination of observation and transition disturbances have the effect of linearizing the trend of $H(A^j|S^j)$ for high utility values. The presence of the transition disturbance $\delta$ has the effect of increasing a little the average utility. Furthermore, in Fig 13D, $H(S^j)$ is reduced for large utility values and has minimum of 5.18 bits, which is similar to the minimum of the state entropy (5 bits) reached by the human participants (Fig 4A). In Fig 13E and 13F, we report the action and state entropies for disturbances $\eta$ and $\delta$ with large magnitude ($\sigma_{\theta}^O = 1$, $\sigma_{\hat{\theta}}^O = 2$, $\sigma_{\theta}^T = 1$, $\sigma_{\hat{\theta}}^T = 2$). As expected, when the noise injected in the system is very large, in most of the simulations the pendulums are not balanced at all or balanced for a very short time. In addition, the large noise decouples states and actions, considerably reducing the conditional action entropy $H(A^j|S^j)$ and keeping the state entropy large $H(S^j)$ for all simulations.

The scatter plots reported in Fig 13C and 13D resemble the action and state entropy trends found analyzing the human data. Although the addition of transition disturbances tends to reduce the positive action entropy's Spearman correlation ($\rho = 0.67$, $p < 0.01$), the Pearson correlations of the human experiment ($r = 0.967$) and the artificial controllers ($r = 0.961$, $p < 0.01$) are nicely matched. In addition, the state entropy correlation is $\rho = -0.982$ when the pendulum is controlled by the human participants, whereas it is $\rho = -0.885$ ($p < 0.01$) for the artificial controllers. While the trend of $H(A^j|S^j)$ in Fig 13C is similar to the one found for human participants, we have observed in all the performed simulations that usually, for large utility values, the maximum of $H(A^j|S^j)$ (e.g., 1.17 bits in Fig 12C or 1.34 bits in Fig 13C) is larger than the maximum $H(A^j|S^j)$ observed in the human experiment (0.78 bits in Fig 2A). We believe that this may be due to the tendency of the artificial controllers to change their actions more frequently or, in other words, to the inclination of the participants to choose the action $n$ (i.e., no-action) more often. Finally, since we have seen that the data points of $H(S^j)$ can be fit by a function shaped as a "half-parabola" in both the run simulations (when the noise is not too large) and in the human experiment, we believe that this trend may be independent from the employed control strategy and rather be a consequence of the relationship between the adopted reward function and the pendulum dynamics.

## 5 Discussion and conclusions

In this study, we asked whether low entropy of action or state distributions could be considered as signatures of mastery of a pendulum balancing task [28, 46, 47]. Our results indicate that for inverted pole control skilled participants keep the variability of the *states* they occupy under control, whereas this variability is higher in less skilled participants (Figs 4 and 5). In other words, our results indicate that skilled participants actively reduce state information to perform efficient control—with this compression of the state space indexed by $H(S^i)$, which is

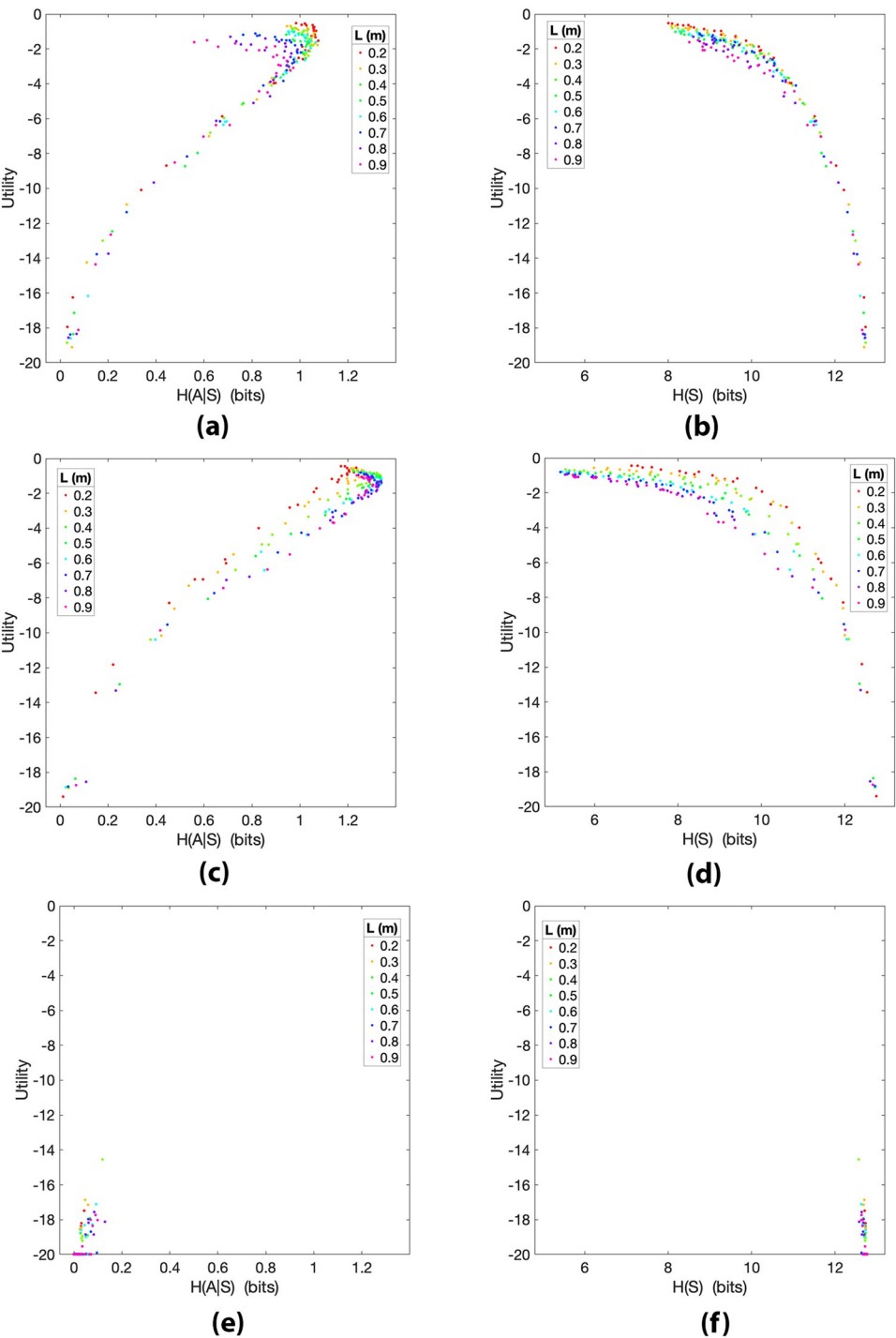

**Fig 13. Utility $U^j$ of simulations $j$ plotted as a function of $H(A^j|S^j)$ ((a,c,e)) and $H(S^j)$ ((b,d,f)) for all pendulum lengths $L$ and artificial controllers with different observation disturbances $\eta$ and transition disturbances $\delta$. (a)** Observation standard deviations $\sigma_\theta^O = 0.01$, $\sigma_{\dot\theta}^O = 0.07$; transition standard deviations $\sigma_\theta^T = \sigma_{\dot\theta}^T = 0$; Spearman correlation $\rho = 0.694$ ($p < 0.01$). **(b)** $\sigma_\theta^O = 0.01$, $\sigma_{\dot\theta}^O = 0.07$; $\sigma_\theta^T = \sigma_{\dot\theta}^T = 0$; $\rho = -0.957$ ($p < 0.01$). **(c)** $\sigma_\theta^O = 0.01$, $\sigma_{\dot\theta}^O = 0.07$; $\sigma_\theta^T = 0.007$, $\sigma_{\dot\theta}^T = 0.014$; $\rho = 0.67$ ($p < 0.01$), Pearson correlation $r = 0.961$ ($p < 0.01$). **(d)** $\sigma_\theta^O = 0.01$, $\sigma_{\dot\theta}^O = 0.07$; $\sigma_\theta^T = 0.007$, $\sigma_{\dot\theta}^T = 0.014$; $\rho = -0.885$ ($p < 0.01$). **(e)** $\sigma_\theta^O = 1$, $\sigma_{\dot\theta}^O = 2$; $\sigma_\theta^T = 1$, $\sigma_{\dot\theta}^T = 2$; $\rho = 0.582$ ($p < 0.01$). **(f)** $\sigma_\theta^O = 1$, $\sigma_{\dot\theta}^O = 2$; $\sigma_\theta^T = 1$, $\sigma_{\dot\theta}^T = 2$; $\rho = -0.574$ ($p < 0.01$).

lower for higher levels of utility. On the contrary, the variability of *actions* during the control tasks is higher for skilled participants (Fig 2A). Our results are in keeping with perceptual control theory and related accounts [16–18]. These suggest that accurate motor control requires reducing the variability of states, including those that are outcomes of the chosen actions—whereas this state compression can be achieved by variable means (hence corresponding to a high variability of actions). Our results are instead incompatible with theories that associate skill mastery with the routinization of action and the decrease of their variability [48, 49].

An important issue here is the task-specific amount of variability that controllers inject into the system. The compensation a skilled controller has to perform might be a driving factor behind the level of action entropy. In our task, skilled participants spend more time in the balancing regions (Fig 8), which require more fine-grained control and motor variability, leading to more action entropy. This is a feature of a range of control problems, where motor variability rather than being only important during early phases of learning due to exploration, remains high also during the successful execution of challenging tasks [9–13]. In this regard, the classification of variability sources in advanced stages of motor learning of redundant tasks proposed in [50] suggests an interesting unifying framework. In this classification, tasks can be characterized by a decrease of action variability when noise reduction plays a major role in successful performance. Alternatively, they can be characterized by both a decrease of state variability and a large actions dispersion when action variables need to co-vary to increase invariance and accuracy in the state space. This may help explaining why in the case of balancing an inverted pendulum we found that the action variability of skilled participants increases rather than decreases. A further study in this direction is left for future investigations.

In general, it is not easy to disambiguate how much of the measured variability and resulting entropies is a result of a given task structure and how much is a result of noise introduced by the controller. One way to disambiguate between these different variability sources is to compare the human data with data generated by a synthetic controller, such as the MDP controller used in our study. We chose to computationally model the pendulum balancing task with a standard Markov decision process for the sake of simplicity and reproducibility (the readers interested in the implementation of a PCT-based model for inverted pendulum control may have a look at [19]). For the studied pendulum task, the human data shows that the observed action variability can not be fully explained by a deterministic controller. To add random disturbances, both in forms of observation and transition noise, creates behaviour that is a much closer fit, and shows similar relationships to the observed empirical data, indicating that the measured action variability originates from the specific participants' control strategy. Fig 11A shows that in principle multiple solutions of the pendulum balancing task are possible for large utility values (similarly, this is shown in Figs 12A and 13A). Some of these solutions have large action entropy, others (in the upper left portion of the plot) have small action entropy. On the contrary, our results show that the solutions adopted by the biological controllers do not span this large set of strategies, but only a subset of the possible solutions (i.e., those with large action entropy for large utility). This phenomenon could be explained by the fact that the human controllers operate under sensory uncertainty over the state of the pendulum and its dynamics, as shown by the artificial controllers in Sections 4.3.2 and 4.3.3 respectively. In favour of this interpretation, Fig 13 contains the plots that better reproduce the trend of the human action and state entropies (compare Figs 13C with 2A and 13D with 4A, respectively). An alternative interpretation of our results is that the action entropy plots are simply a necessary consequence of the structure of the pendulum task, and it would be impossible to design a controller that formalizes the routinization hypothesis. However, we performed a control simulation (see S1 File) showing that a controller that formalizes the routinization hypothesis is

possible and hence our results indeed reflect characteristics of the human controller and not just of the structure of the task.

The fact that in all the performed simulations we found a similar monotonic decreasing trend for the state entropy, seems to show that the observed state variability comes from the relationship between the employed utility measure and the pendulum dynamics. As a further caveat, we need to consider that it is possible that more than one mechanism is responsible for the state compression that we report here. Firstly, to balance the pendulum its trajectories must be squeezed into the right part of phase space. To obtain such entropy reduction the needed information [51] needs to be reflected in the actions. Then, at the level of representation, experts could abstract the state space to use less information to codify the task, encoding some portion of the state space more precisely and other portions in a more coarse manner. In the experiment, a more compact representation of the state space for more skilled participants is indicated by the number of effectively visited states. This was confirmed by our analysis, where more skilled participants visited a smaller portion of the phase space (compare Fig 7A and 7B with 7C and 7D). To establish whether experts would be better than non-experts in appropriately coding the state space is left to future work, together with determining which resulting latent space these may use to efficiently balance the pendulum. The problem of identifying which of the pendulum's dimensions may be more relevant for the control task is addressed by the "uncontrolled manifold" hypothesis [52], which states that irrelevant dimensions usually remain uncontrolled by agents.

Behind the reduction of $H(S^i)$ and the increase of $H(A^i|S^i)$ there is the need of controlling the state at the expense of increasing action variability to perform efficient control. However, it is important to distinguish between controlled and uncontrolled variability. For the case of controlled variability, experts can actively choose to be variable (as it happened for $A^i$ in Fig 2A); or they can choose to minimize the variability of the future trajectory (as it happened with S and $S'^i$ in Figs 4 and 5). However, there is a component of human control that is not part of the participants intentions and that is injected in the pendulum dynamics and effectively becomes part of the uncontrolled variability of the system. All the system's uncertainty is ultimately generated by the participants, because the pendulum dynamics is deterministic and almost conservative. So, uncertainty in control is essentially translated by the pendulum in uncertainty of its dynamics, which in the end is transformed in the uncontrolled variability of the state. But the entropy $H(S^i)$ alone does not allow us to distinguish between the controlled and uncontrolled variabilities of the state and ways to detect this difference will be investigated in future work

Another important objective for future research is understanding the generality of the findings reported in this study. We reported that while performing a particular kind of skilled behaviour—here, the control of an inverted pendulum—people show low entropy of states but high entropy of actions. However, it remains to be understood to what extent this result extends to other kinds of skilled behavior that may have different characteristics and whether the prevalence of low entropy of states or of actions is task-dependent. A large body of research has shown that when learning and performing skills that require the manipulation of objects, people use sophisticated strategies; for example, they select appropriate initial conditions for the control task that simplify subsequent interactions [53] and make the object dynamics predictable [54]. Moreover, in keeping with classical work of Bernstein [55], recent studies found that people and animals modulate various aspects of the variability of their actions and sensations during learning in adaptive ways; for example, they counteract the maladaptive effects of noise by channeling it into task-irrelevant dimensions and increase their movement variability when their performance is poor [50, 56–58]. Understanding which specific task characteristics

cause the low entropy of states reported in this study and the more sophisticated strategies reported above is an open objective for further investigations.

The compelling structure of the reported utility-entropy curves and the clear correlations that emerged from our analyses prompt the question of where the nature of this trade-off comes from. In general, we expect that the specific nature of the task is important—and that the trade-offs that emerged from our analysis could be especially compelling in tasks which require reaching well defined (goal) states. For example, in the pendulum task, the final goal state distribution must have low entropy by definition (e.g., the correct position of the pendulum is almost unique). Especially in difficult tasks, there might be a very limited number of ways to achieve these goal states, forcing (skilled) participants to reduce the entropy of their trajectories to the goal states—or in other words, forcing them to pass through a "narrow corridor" of states to achieve the desired goals. For example, some aspects of skilled behavior, such as perfecting serving in tennis in order to put the ball in a specific location, requires a long training that could lead to low-variance, stereotyped movements. Other tasks, have fewer constraints to fulfill in order to be completed, such as simply putting the ball over the net. Usually, goals with high state variability can be achieved by many possible trajectories, implying large state and action entropy. A characterization in this regard would be to look at the entropy of the task's goal state as possible factor determining state and action entropy for skilled behavior (e.g., in cooperative vs. antagonistic tasks). Nevertheless, this characterization would not be complete. In fact, although the pendulum balancing task has small goal entropy, and few specific trajectories must be traversed in order to reach it, the actions entropy for skilled behavior is large rather than small. Since its goal is an unstable state, once reached, this needs to be sustained, requiring large action entropy. Hence, when there are random disturbances in the control system, also the stability of the goal may determine the action entropy of skilled behavior. Therefore, it is likely that in the balance between entropy generation by the system, vs. by the agent's actions and/or observations, all play some role in determining the precise form of interplay between state and action variability and task mastery.

In summary, in this study we observed the relationship between skilled performance and low entropy of states and high entropy of actions for the inverted pendulum balancing task. If these relationships generalize to other tasks is subject to further study. We do provide a synthetic model which can, with the modeling of observation and transition noise, produce similar-looking information-theoretic measures, but we recognize that even in this seemingly simple task one can find alternative explanations for how those relationships could be generated.

## Supporting information

**S1 File. SupplementaryMaterial.pdf.** This document contains an additional numerical study based on a computational model of information-theoretic bounded rationality (i.e., the Relevant Information formalism) that supports the routinization hypothesis for the pendulum balancing task presented in this paper. The reported relevant information policy has low action entropy for high levels of skill. This shows that in principle the routinization hypothesis, although confuted by our study on human motor control, has a certain degree of plausibility for the pendulum balancing task, highlighting the biological significance of our findings. (PDF)

**S2 File. ExperimentalData.zip.** The dataset within the file contains the data collected during the experiment. Each CSV file named "FRI_p_l.csv" corresponds to a different trial, where "p" stands for an unique numerical participant's identifier and "l" stands for pendulum length (where "2" corresponds to 0.2, "3" to 0.3 m, "4" to 0.4 m, etc.). In every file, rows correspond

to subsequent time steps, sampled with a frequency of 0.017 s. For every recorded time step, each column contains:

1. the action selected by the trial's participant ("action"). The value 0 corresponds to a "no action", the value 1 represents an action that pushes the pendulum towards the clockwise direction and the value 2 corresponds to an action that pushes the pendulum towards the anti-clockwise direction.

2. the angular velocity of the pendulum at that time step ("ang. velocity"), measured in rad s$^{-1}$.

3. the angle of the pendulum at that time step ("angle"), measured in rad with values in the interval $[0, 2\pi)$ rad. The angle 0 rad corresponds to the pendulum oriented in its upright position.

4. the single utility value associated to the whole trial ("utility").
(ZIP)

**S3 File. InfoAnalysisResults.csv.** The file contains the results of the information-theoretic analysis of the aforementioned data. Each row of the spreadsheet represents a different trial. For each trial, the columns contain:

1. pendulum length ("Pend. Length").

2. trial's utility ("Utility").

3. entropy of the state ("H(S)").

4. conditional entropy of action given state ("H(A|S)").

5. conditional entropy of next state given action ("H(S'|A)").

6. mutual information between action and next state ("I(A;S')").
(CSV)

## Author Contributions

**Conceptualization:** Nicola Catenacci Volpi, Martin Greaves, Dari Trendafilov, Christoph Salge, Giovanni Pezzulo, Daniel Polani.

**Data curation:** Nicola Catenacci Volpi, Martin Greaves, Christoph Salge.

**Formal analysis:** Nicola Catenacci Volpi, Dari Trendafilov, Christoph Salge, Daniel Polani.

**Funding acquisition:** Giovanni Pezzulo, Daniel Polani.

**Investigation:** Nicola Catenacci Volpi, Martin Greaves, Dari Trendafilov, Christoph Salge, Giovanni Pezzulo, Daniel Polani.

**Methodology:** Nicola Catenacci Volpi, Martin Greaves, Dari Trendafilov, Christoph Salge, Daniel Polani.

**Project administration:** Martin Greaves, Daniel Polani.

**Resources:** Nicola Catenacci Volpi, Martin Greaves, Dari Trendafilov, Christoph Salge.

**Software:** Nicola Catenacci Volpi, Dari Trendafilov, Christoph Salge.

**Supervision:** Giovanni Pezzulo, Daniel Polani.

**Validation:** Nicola Catenacci Volpi, Dari Trendafilov, Giovanni Pezzulo, Daniel Polani.

**Visualization:** Nicola Catenacci Volpi, Martin Greaves, Dari Trendafilov.

**Writing – original draft:** Nicola Catenacci Volpi, Giovanni Pezzulo.

**Writing – review & editing:** Nicola Catenacci Volpi, Dari Trendafilov, Christoph Salge, Giovanni Pezzulo, Daniel Polani.

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
