## [Decision Letter · Decision Letter 0]

31 Mar 2022

Dear Dr. Catenacci Volpi,

Thank you very much for submitting your manuscript "Skilled motor control implies low entropy of states but high entropy of actions: an information-theoretic study of perceptual control versus routinisation" for consideration at PLOS Computational Biology.

As with all papers reviewed by the journal, your manuscript was reviewed by members of the editorial board and by several independent reviewers. In light of the reviews (below this email), we would like to invite the resubmission of a significantly-revised version that takes into account the reviewers' comments. To summarize the main points in these comments: (1) More development and comparison of specific models is needed to make a compelling case for the theoretical account; (2) How general are the conclusions we can draw from the pendulum task?

We cannot make any decision about publication until we have seen the revised manuscript and your response to the reviewers' comments. Your revised manuscript is also likely to be sent to reviewers for further evaluation.

Sincerely,

Samuel J. Gershman

Deputy Editor

PLOS Computational Biology

Reviewer's Responses to Questions

**Comments to the Authors:**

Reviewer #1: In their research paper „Skilled motor control implies low entropy of states but high entropy of actions: an information-theoretic study of perceptual control versus routinisation” the authors investigate an inverted pendulum balancing task with human subjects for different pendulum lengths (ranging from short – difficult – to long – easy). They find a positive correlation between task success (utility) and the entropy over actions and a negative correlation between task success and the entropy over states. They conclude that their results argue in favour of perceptual control (low entropy over states) and against learning by routinization (low action entropy).

I am very sympathetic to the approach of the paper, although I think several points need to be clarified and amended before the study can be published.

The title suggests that low state entropy and high action entropy are always implied by skilled motor control and hence serve as a general principle. However, this is most likely not always the case, as the authors themselves admit in their discussion when citing previous studies that have investigated different kinds of variability modulation during motor learning including noise reduction [e.g. ref 41]. This is also foreshadowed in the classical work of Nicolai Bernstein who proposed already in the 60s after studying the degrees of freedom problem that motor variability is modulated adaptively depending on the stage of learning. Thus, the story of skilled motor behaviour is most probably more complex than simply minimizing state entropy. This could not only be reflected in the title, but also discussed in more detail, for example by applying entropy concepts to the skittles game studied in 41 or something similar.

Also, the analysis does not clarify the generality of the proposed principle, i.e. in how far the results depend on the particular task chosen. It seems that small and large pendulum sizes are somewhat similar to different cursor gains on a computer screen, where large gains induce more variability by magnifying natural motor noise. Could the results just be a consequence of the fact that smaller pendulum sizes lead to the induction of more motor noise into the system? This and similar questions could be investigated by simulation studies with learning controllers that have different noise properties. The authors write that simulation studies are left for the future, but I can see no reason not to include them in the study (many different learning algorithms exist for the inverted pendulum) as they seem essential to judge the generality of the results (also it is pretty standard to include modeling in PLoS Comp Biol).

Minor comments:

Methods:

* how many male/female participants? Age range?

* why is Euler integration used instead of a more precise integration method? The pendulum equation is known to have significant integration errors (the undamped pendulum simulated by Euler increases in energy indefinitely). Can integration errors be neglected?

* in equation (5) and the other equations is the entropy computed for a single trial (i)? I would have thought the entropy over all state trajectories? Similarly, what is the probability P(s^i) and how is it computed? I would have thought it is the probability of a single state trajectory? But they would be continuous, which means that entropy would not be defined (unless it’s differential)? How would this distribution be estimated from finitely many samples? Can the trajectories be thought of as Markov processes and be quantified by entropy rates? The authors reference a toolbox, but these questions should be clarified in the paper, what options exist to quantify entropy for this task etc.

* rather than comparing different lengths of pendula, would it not be good to (also) study entropy values in early and late learning for the same pendulum length? Skilled motor behaviour would correspond to late learning and the entropy values could be compared to unskilled early learning.

Reviewer #2: There is a strong justification for this research study, considering the continuing assumption that actions are learned, becoming more stable over time. There have been many authors in recent years who have drawn on perceptual control theory to point to the likely flaw in this logic owing to the necessity of behavioural variability to establish and maintain control. The design of the study is to be praised too, providing an ecologically relevant and classic inverted pendulum task to human participants and coding movement data continuously and accurately, in addition to the analytic methods employed, although in my opinion they could have been a lot simpler to illustrate the same point without an information-theoretic standpoint. Nonetheless, this study provides a valuable point of contact between information theory and PCT for later research.

My main issues regard the relative lack of detail regarding how perceptual control systems actually work, and in turn a lack of a clear and simple explanation for how these findings were obtained so consistently and clearly.

In my opinion, the article requires the reproduction of a closed loop PCT model of how an inverted pendulum might be maintained at a vertical state. There are published examples available that use a three of four level hierarchy, but this could be simplified if made explicit. The important points, though are threefold:

1) Actions in PCT are the control of perceptual input. Therefore, when a human participant moves their arm hand in a certain direction to keep an inverted pendulum stabilised, this itself involves the control of variables that are hierarchically subordinate to the control of the position of the bob. These may not be the focus of the study, but in order to construct a model (or a robot; see Johnson et al., 2020) to balance an inverted pendulum, control of these variables need to be included. The definition of action/behaviour as the control of input in PCT needs to be clear in this manuscript.

2) Following PCT, there would be two reasons for the variability in recorded actions in this study, and the diagram above would be necessary to explain them. The first reason is that actions counteract unmeasured disturbances to the movement of the bob, which or often from the side effects of the controllers own movements. These counteracting actions seem to be 'variable' but this is only because the disturbances are not measured; in fact they are fairly deterministic consequences of the error from the vertical reference point caused by a disturbance unfolding over time. This is clearly demonstrated by the close correlation between disturbance and action in tracking tasks (for a review, see Parker, 2020).

The second reason for variability in action will be the effect of truly random trial-and-error changes to the parameters of the human control system (e.g. the gain, input functions) that develop over time during the learning of the task. As far as I can tell, currently, the article does not clarify these distinctions.

3) The study provides some very convincing tests of the hypotheses made. However, ultimately the test of a theory is whether it enables the construction of models (or robots) that replicate the naturalistic phenomenon they purport to explain. This is how we appreciate the 'truth' of the theories within physics to enable advances in technology, for example. Therefore, whilst this study provides good evidence to support PCT, it does not test the principles of PCT in detail. For further discussion of this issue see Huddy & Mansell (2018).

Overall, this is a very welcome advance in research methodology to complement the emerging empirical literature supporting PCT, that needs some greater detail and clarity regarding this theory and how it actually 'works'.

Reviewer #3: This paper asks the question of whether skilled motor control is more consistent with with perceptual control (reducing variance of states) or “routinization” (reducing variance of actions). The authors design an experiment where participants have to control an unstable pendulum via keypresses to achieve stability. They predict that skilled control of this dynamical system should lead to low levels of state entropy but high-action entropy. They confirm their hypothesis in human experiments, showing a greater variability in key presses (high action entropy) and lower variability in the state space when the pendulum is successfully controlled. Additionally, they show that controlling a longer pendulum is easier than a shorter pendulum, yet requires more key presses in the balanced region to maintain a steady-state. I do think the information-theoretic analyses are interesting, especially when it comes to the higher vs. lower complexity control tasks (short vs. long pendulums).

Major comments:

- One overarching flaw I find in this paper is in how the background is set up. In the Introduction, the authors talked about wanting to understand mastery of skilled behavior, but then quickly launch into two opposing predictions: perceptual control theory, where the goal is to reduce the variance in the states, versus routinization, which is where the variability of actions is reduced to achieve a highly practiced motor skill. However, I don't find that these are two valid opposing hypotheses, because it seems clear and intuitive that the type of control will depend entirely on the task at hand. For example, in the author summary the others talked about skills like tennis. But tennis is really different than the task in their actual experiment---to perfect a serve in tennis, one has to practice and repeat the serve over and over again to achieve low-variance, stereotyped movements. Yet, in the pendulum balancing task, the goal is to maintain a stable state, and the way to do that is to constantly vary your actions. In lines 380-384, the authors even admit that this finding (of action entropy being higher in the task) was quite obviously true from the beginning given the task, rendering the exposition of these two “opposing” hypotheses obsolete.

- Because “state” can often be defined in various ways, one could argue that in the tennis example, the steady-state that is being maintained is the successful serve of the ball. The tennis player must find an action that ensures that the ball will make it over the net every time. The main difference in maintaining this steady state versus the steady state of a pendulum is that while there could be many different actions (e.g., serves) that are successful, it is often most advantageous to perfect one type of serve that works every time, instead of trying out a different serve for each game that is played. But an additional question is whether the final goal state distribution has a high or low entropy: does it matter where exactly the ball lands on the other side, or is a serve successful as long as the ball goes over the net?

- I would suggest that the authors rewrite the introduction to be more specific about their opposing hypotheses, because the current setup makes the low action entropy hypothesis seem like a straw man especially when the particular task is being introduced. This might be a more interesting hypothesis if there were several tasks that were being studied which varied the entropy of the final goal state.

- Finally, it might be too far-reaching of a claim to say that your results apply to “skilled action control” in general…

- In section 5.3, I suggest reframing the motivation for doing an analysis of time spent in the balancing region. What question are you trying to answer here?

Minor comments / questions:

- 223: why did you bin state space into different sizes? 1000 bins for angle interval vs 200 bins for angular speed interval?

- 251: why 270ms after?

- I would recommend citing some more recently work in the motor skill learning literature. See more work by Bence Olveczky’s lab, such as Dhawale et al. 2019, Current Biology. In their study, animals vary their actions to maintain the steady-state of reward, and change their actions when they notice that the reward region has changed.

**Have the authors made all data and (if applicable) computational code underlying the findings in their manuscript fully available?**

Reviewer #1: Yes

Reviewer #2: Yes

Reviewer #3: Yes

PLOS authors have the option to publish the peer review history of their article (what does this mean?). If published, this will include your full peer review and any attached files.

Reviewer #1: No

Reviewer #2: **Yes: **Warren Mansell

Reviewer #3: No
---

## [Decision Letter · Decision Letter 1]

12 Sep 2022

Dear Dr. Catenacci Volpi,

Thank you very much for submitting your manuscript "Skilled motor control of an inverted pendulum implies low entropy of states but high entropy of actions" for consideration at PLOS Computational Biology. As with all papers reviewed by the journal, your manuscript was reviewed by members of the editorial board and by several independent reviewers. The reviewers appreciated the attention to an important topic. Based on the reviews, we are likely to accept this manuscript for publication, providing that you modify the manuscript according to the review recommendations. In particular, R1 believes that you need to more clearly distinguish between the two hypotheses, suggesting that you do this through the lens of learning. I'm not sure you need to study learning in this setting, if you can come up with another way to make the point.

Sincerely,

Samuel J. Gershman

Section Editor

PLOS Computational Biology

[LINK]

Reviewer's Responses to Questions

**Comments to the Authors:**

Reviewer #1: In their revised manuscript the authors have added an MDP model of the task, which is great, but they do not study learning, nor do they investigate / report the effect of pendulum length, nor do they produce quantitative predictions under the two hypotheses they consider. The main problem I see is the following:

the authors motivate the study with wanting to distinguish whether motor skilll is rather reflected by routinization (signature: small action entropy) or perceptual control (signature: small state entropy). it seems intuitive that any kind of feedback process (like tracking, pednulum control, etc.) should probably follow the latter. however, the question is what is small and what is large in terms of entropy? the natural answer, it seems to me, would be to study the learning process: which entropy is reduced over time? but instead the authors study different lengths of pendula. so the question is, how does one pendulum length compare with another in terms of the two hypotheses, and what do the results shown in Figure 3 actually mean? does it simply mean: controlling a smaller pendulum is more difficult, therefore it is associated with lower utility and higher state entropy. but is this just a consequence of feeding the same amount of noise into different pendula with different lengths? how would that distinguish between the two different hypotheses from the beginning? this would suggest that the entropy plots are simply a property of the task and do not tell us much about the human controller. is there even a controller conceivable in this task that would formalize the routinization hypothesis? it has to be clear somehow that the results tell us something about the biology and not about inverted pendula with different lengths fed by some level of noise. in my mind the cleanest way would be to formalize systematically controllers corresponding to the two hypotheses and study their behavior with pendula of different lengths and confirm that they make different predictions, and that human data fits one of them and not the other (a model-based quantitative comparison). but maybe instead the authors want to study learning and show reduction in entropy (and maybe use reinforcement learning models to distinguish between the two hypotheses) so the question about the different lengths of pendula does not arise, or maybe there is another argument one could make.

minor comment:

figure 9 and 10 need the correct lengths in the legend.

Reviewer #3: The authors have addressed all my comments in a clear and concise manner and have integrated respective changes. I appreciated the discussion in the reply to reviewers about cooperative vs antagonistic task goals. The introduction is now much more clear as to the scope and question of the paper, and the authors are clear about the generalizability of the findings in the Discussion. I can recommend this paper for publication!

**Have the authors made all data and (if applicable) computational code underlying the findings in their manuscript fully available?**

Reviewer #1: Yes

Reviewer #3: Yes

PLOS authors have the option to publish the peer review history of their article (what does this mean?). If published, this will include your full peer review and any attached files.

Reviewer #1: No

Reviewer #3: **Yes: **Lucy Lai

Figure Files:

Data Requirements:

Reproducibility:

References:

---

## [Editor Report · Decision Letter 2]

12 Dec 2022

Dear Dr. Catenacci Volpi,

We are pleased to inform you that your manuscript 'Skilled motor control of an inverted pendulum implies low entropy of states but high entropy of actions' has been provisionally accepted for publication in PLOS Computational Biology.

Best regards,

Samuel J. Gershman

Section Editor

PLOS Computational Biology

---

## [Editor Report · Acceptance letter]

29 Dec 2022

PCOMPBIOL-D-22-00262R2 

Skilled motor control of an inverted pendulum implies low entropy of states but high entropy of actions

Dear Dr Catenacci Volpi,

I am pleased to inform you that your manuscript has been formally accepted for publication in PLOS Computational Biology. Your manuscript is now with our production department and you will be notified of the publication date in due course.

With kind regards,

Zsofia Freund
